# EPSVEC: Efficient and Private Synthetic Data Generation via Dataset Vectors

Amin Banayeeanzade [* 1]   Qingchuan Yang [* 1]   Deqing Fu [1]   Spencer Hong [2]   Erin Babinsky [2]
Alfy Samuel [2]   Anoop Kumar [2]   Robin Jia [1]   Sai Praneeth Karimireddy [1]

## Abstract

High-quality data is essential for modern machine learning, yet many valuable corpora are sensitive and cannot be freely shared. Synthetic data offers a practical substitute for downstream development, and large language models (LLMs) have emerged as powerful engines for generating it. However, existing private text generation methods are severely inefficient: they are data-intensive, computationally slow, and often require large private corpora or batch sizes to achieve usable quality. We introduce EPSVEC, a differentially-private lightweight alternative that steers LLM generation using *dataset vectors*–directions in activation space that capture the distributional gap between private data and public priors. EPSVEC extracts and sanitizes steering vectors just once and then performs standard decoding. This decouples the privacy budget from generation, enabling arbitrarily many synthetic samples without additional privacy cost and yielding strong fidelity even in low-data regimes. Furthermore, we enhance our method by utilizing pretrained (base) models and introducing fixed-shot prompting to boost generation diversity and fidelity. Our experiments demonstrate that EPSVEC outperforms existing baselines in distributional alignment and downstream utility, particularly in low-data regimes, while significantly reducing computational overhead.

## 1 Introduction

Modern machine learning systems are becoming increasingly data-intensive. Yet the most valuable text corpora, such as biomedical records, internal documents, and user

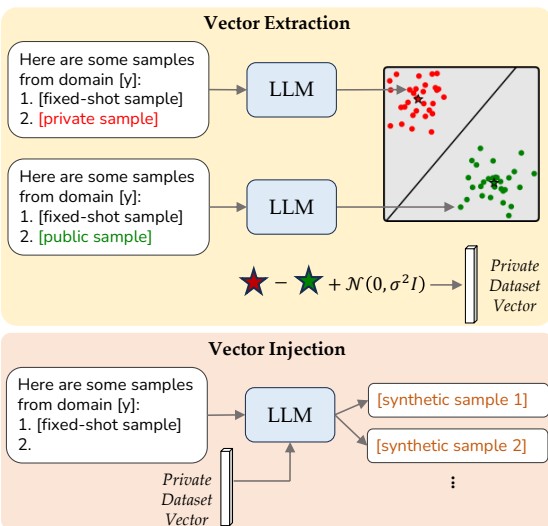

*Figure 1.* Overview of our method. We extract dataset vectors by distilling the private dataset into a compressed vector. Protected with enough noise, this vector can be injected into the hidden states of the LLM at inference time to efficiently generate any desired amount of synthetic data while ensuring differential privacy.

feedback, are often inherently sensitive, limiting their use in model development and evaluation. In parallel, Large Language Models (LLMs) have emerged as powerful engines for synthetic data generation, in particular when real data are scarce, restricted, or costly to share (Ponomareva et al., 2025). Naive generation strategies (e.g., zero-shot or few-shot prompting) frequently fail to match the target distribution or can inadvertently leak sensitive content, resulting in synthetic data that are either low-fidelity or insufficiently private (Choi et al., 2025). Training-based approaches such as private fine-tuning (Carranza et al., 2024; Kurakin et al., 2024; Yu et al., 2024; Mattern et al., 2022) are data-hungry and computationally expensive, while inference-time private prediction methods (Amin et al., 2024; 2025) incur substantial privacy costs during generation—often scaling with the number of produced tokens.

In many realistic settings, practitioners need a method that (i) works in low-data regimes, (ii) is lightweight in compute, and (iii) supports large-scale sampling without privacy risks. These challenges motivate a central question:

*How can we efficiently generate high-quality synthetic data with rigorous privacy guarantees?*

---

[*]Equal contribution [1]Thomas Lord Department of Computer Science, University of Southern California [2]Capital One. Correspondence to: Amin Banayeeanzade <banayeea@usc.edu>, Qingchuan Yang <qcyang@usc.edu>.

🞅 Code at: Github.com/noiseeboi/EPSVec.

*Proceedings of the $43^{rd}$ International Conference on Machine Learning*, Seoul, South Korea. PMLR 306, 2026. Copyright 2026 by the author(s).

In this work, we propose **E**fficient and **P**rivate **S**ynthetic data generation via dataset **Vec**tors (EPSVEC). At the core of EPSVEC are *dataset vectors*: single directions in an LLM's embedding space that compactly encode a dataset's characteristic properties. Beyond coarse attributes such as topic or sentiment, an LLM's embedding space captures higher-order structure—including stylistic conventions, subtopic mixtures, and semantic flow—that is difficult to specify through natural-language descriptions alone (Tennenholtz et al., 2024; Simhi & Markovitch, 2023). EPSVEC extracts dataset vectors from private-data embeddings and then releases a noisy version satisfying Differential Privacy (DP). At generation time, we inject the privatized dataset vector into the model's hidden states to steer decoding toward the target dataset attributes (see Figure 1).

Dataset vectors are computed and privatized once, and then reused for all downstream generations. After this one-time release, all subsequent steps—including decoding, filtering, and other post-processing—*incur no additional privacy cost*. This yields several practical advantages: (i) EPSVEC can generate an arbitrary number of synthetic samples; (ii) samples may be of arbitrary length, without per-token privacy costs; (iii) constructing dataset vectors does not require large amounts of private data; and (iv) generation is compute-efficient with the same cost as zero-shot generation.

Beyond EPSVEC's core dataset-vector component, we introduce a practical technique for reliably leveraging *pretrained-only* (base) LLMs for private data generation. Base models retain broader support and stylistic diversity than instruction-tuned variants (Shypula et al., 2025), which is crucial for achieving high fidelity to the private data distribution. However, they are harder to control with zero-shot prompts and can drift in format and coarse attributes. We therefore utilize private histograms to select a small set of fixed-shot exemplars and reuse them as a prompt scaffold throughout the pipeline. Our fixed-shots stabilize prompting and improve fidelity, while preserving formal privacy guarantees.

Experiments show that EPSVEC delivers large gains in distributional fidelity, as measured by MAUVE (Pillutla et al., 2021), especially in low-data regimes. Notably, it achieves an average 150% MAUVE improvement across 4 datasets over the next-best method (Amin et al., 2025). Moreover, EPSVEC maintains comparable quality to real data even under stronger privacy guarantees where previous methods undergo severe quality trade-offs. When used for BERT finetuning, synthetic data from EPSVEC achieves comparable accuracies with real data.

## 2 Related Work

**Steering and representation engineering.** Controlling LLM behavior has evolved from prompt engineering to direct interventions in the model's internal representations (Zou et al., 2025; Banayeeanzade et al., 2025; Gan et al., 2025). Most relevant to our work are activation steering approaches that intervene on hidden states at inference time. Subramani et al. (2022) showed that latent steering vectors can deterministically shift generation targets. Turner et al. (2024) demonstrated that simple linear activation additions, computed from contrasting prompt pairs, reliably toggle model behaviors without optimization.

Directly relevant to our work, PSA (Goel et al., 2025) studies differentially private activation editing for LLM alignment, constructing steering vectors from paired positive/negative demonstrations. In contrast, EPSVEC targets *private synthetic data generation*: we learn dataset vectors that encode *dataset shift*—the direction from a matched synthetic reference distribution toward the real private corpus—rather than demonstration-based behavioral edits. We further introduce a DP *fixed-shot* scaffold to improve the reference distribution and stabilize base-model generation.

**DP synthetic data generation.** In the text generation domain with LLMs, prior work mainly falls into two families (Ponomareva et al., 2025): The first family uses DP training/finetuning, where a pretrained LLM is adapted to the private corpus under DP-SGD (Yu et al., 2022; Hong et al., 2024; Liu et al., 2025). These approaches can achieve strong fidelity, but require difficult and expensive DP optimization.

The second family explicitly avoids DP training and instead enforces DP at inference time. For example, AUG-PE (Xie et al., 2024) prompts models to generate a corpus of synthetic data, followed by private filtering and textual evolution to improve fidelity. Another example is private prediction (PP), where multiple contexts produce next-token distributions that are then privately aggregated (Ginart et al., 2022; Flemings et al., 2024; Cohen et al., 2025).

More recent DP inference methods leverage in-context learning by distributing private examples across many prompts and applying DP token selection at each decoding step (Wu et al., 2024; Duan et al., 2023). Follow-up work improves scalability and utility through parallel composition (Amin et al., 2024; 2025), adaptive clipping (Gao et al., 2025), and better token sampling mechanisms (Vinod et al., 2025). While training-free methods substantially reduce engineering overhead, many of these methods rely on large batches of private data as input and are often compute and data-hungry, especially when strong privacy is required.

## 3 Preliminaries

**Problem definition.** We consider the problem of private synthetic data generation in the text domain. Let $\mathcal{D}_{\text{priv}} = \{(x_1, y_1), \ldots, (x_m, y_m)\}$ be a private dataset of $m$ textual records with their corresponding labels, containing sensitive information that must be protected from direct disclosure. Our goal is to design a randomized mechanism that outputs a *synthetic dataset* $\mathcal{D}_{\text{syn}}$ that is distributionally similar to $\mathcal{D}_{\text{priv}}$, while ensuring that no individual record in $\mathcal{D}_{\text{priv}}$ can

be inferred from the output. Synthetic data should preserve the statistical and semantic properties of the original data and be suitable for downstream tasks. For formal privacy guarantees, we use the Differential Privacy (DP) framework.

**Definition 3.1** (Dwork et al., 2006). A randomized mechanism $\mathcal{M}$ satisfies $(\varepsilon, \delta)$-differential privacy if for any neighboring $\mathcal{D}, \mathcal{D}'$ and any measurable set $\mathcal{O}$ in the range of $\mathcal{M}$,

$$\Pr[\mathcal{M}(\mathcal{D}) \in \mathcal{O}] \ \leq \ e^{\varepsilon} \Pr[\mathcal{M}(\mathcal{D}') \in \mathcal{O}] + \delta.$$

Intuitively, this definition ensures that any single text record in the private dataset has a limited effect on the distribution of any released output, including our DP-protected dataset vectors and any synthetic data generated from them via post-processing.

**Vector representations of dataset shift.** The central abstraction of our method is the *dataset vector*: a fixed-length direction in a language model's activation space that captures the *distributional shift* from a reference text distribution toward a target dataset. Our construction is grounded in the Linear Representation Hypothesis (Park et al., 2024; Rimsky et al., 2024), which posits that semantic concepts and high-level attributes are encoded as linear directions within the activation space of LLMs. We extend this hypothesis to dataset-level properties: if each dataset represents a coherent collection of semantic and stylistic attributes, their subtle differences can be approximated by a characteristic direction aggregated from differences in their samples. Steering the model's generation along this direction then produces predictable changes in generation, moving outputs closer to the target dataset's attributes.

We first empirically validate that dataset-level attributes are explicitly distinguished within the language model's internal representation. We extract embeddings of samples from the multi-topic *BioRxiv* dataset (representing real biological abstracts) from layer 19 of LLAMA-3.1-8B-INSTRUCT. Additionally, we prompt the same model to generate abstracts mimicking BioRxiv's topics and styles. As illustrated in Figure 2, we observe two distinct phenomena in the representation space. First, data samples cluster distinctly by sub-topic (e.g., Bioinformatics and Microbiology), confirming that the model's internal representations are organized by dataset-level attributes. More crucially, there is a clear separation between the *real* samples and the *synthetic* samples. While often hard to articulate by human annotators, the subtle differences between real and synthetic text are salient in the models' representation (Ludwig & Mullainathan, 2023).

These observations motivate the potential of dataset vectors as precise representation-space control signals. Dataset vectors bridge the gap between target data and synthetic data prompted from natural-language, enabling fine-grained control over generation. They allow us to steer the model at generation time toward nuanced characteristics, such as lexical distributions and structural patterns that may be difficult

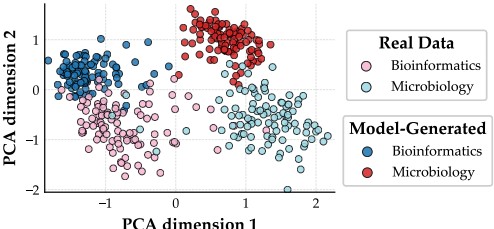

*Figure 2.* First two principal components of BioRxiv abstract embeddings. Model-generated points represent biology paper abstracts generated by zero-shot prompting of LLAMA-3.1-8B-INSTRUCT, while other points show real paper abstracts.

to specify explicitly through natural language instructions.

## 4 EPSVEC: Efficient Generation of Private Synthetic Data with Dataset Vectors

In this section, we present EPSVEC for private synthetic data generation based on dataset vector injection. In the following subsections, we expand on the design of each sub-component. The core mechanism is to utilize private dataset vectors to steer LLM generation at inference time towards the target private distribution (§4.2). Additionally, we motivate the use of pretrained (base) models together with fixed-shot exemplars to improve the quality and diversity of the synthetic samples (§4.3 and §4.4). Finally, we provide an analysis of the overall privacy guarantees of our method (§4.5). Algorithm 1 details the vector extraction steps, and Algorithm 2 presents the full pipeline of EPSVEC.

### 4.1 Extracting Private Dataset Vectors

**Latent representations of data points.** Let $f$ be a frozen LLM with $L$ layers and hidden dimension $d$. For an input text $x$, we run $f$ and extract token-level hidden states at each layer $\ell \in [L]$. We denote the extracted hidden state as $a_{\ell,t}(x) \in \mathbb{R}^d$ for layer $\ell$ and token index $t \in [T(x)]$, where $T(x)$ is the sequence length.

We construct a single example-level representation by mean-pooling over all tokens in the sequence, defined as

$$h_\ell(x) = \frac{1}{T(x)} \sum_{t \in [T(x)]} a_{\ell,t}(x) \in \mathbb{R}^d. \qquad (1)$$

**Mean-difference dataset vector.** Let $\mathcal{D}_y^+ = \{x_i^+\}_{i=1}^n$ be the private dataset with $n$ samples that has the attribute $y$ such that $(x_i^+, y) \in \mathcal{D}_{\text{priv}}$ (e.g., BioRxiv dataset with the Microbiology topic label). We construct a synthetic dataset $\mathcal{D}_y^- = \{x_i^-\}_{i=1}^n$ by prompting the LLM to generate datapoints with attribute $y$.

For a pair of positive and negative samples $(x_i^+, x_i^-)$, line 3 of Algorithm 1 computes the mean difference vector $d_\ell^{(i)}$ between their encoded hidden representations at every layer $\ell$. These difference vectors are then aggregated through

**Algorithm 1** Extract Private Dataset Vectors

**Input:** Private set $\mathcal{D}_y^+ = \{x_i^+\}_{i=1}^n$, synthetic set $\mathcal{D}_y^- = \{x_i^-\}_{i=1}^n$, hidden state extraction $h_\ell(\cdot)$, clip thresholds $\{C_\ell\}_{\ell=1}^L$, and noise scales $\{\sigma_\ell\}_{\ell=1}^L$.
**Output:** Private dataset vectors $\{v_\ell\}_{\ell=1}^L$.

1: **for** $\ell = 1, 2, \ldots, L$ **do**
2:     **for** $i = 1, 2, \ldots, n$ **do**
3:        $d_\ell^{(i)} \leftarrow h_\ell(x_i^+) - h_\ell(x_i^-)$
4:        $d_\ell^{(i)} \leftarrow d_\ell^{(i)} \cdot \min\left(1, \frac{C_\ell}{\|d_\ell^{(i)}\|_2}\right)$
5:     **end for**
6:     $v_\ell \leftarrow \frac{1}{n}\sum_{i=1}^n d_\ell^{(i)} + \xi_\ell$, where $\xi_\ell \sim \mathcal{N}(0, \sigma_\ell^2 I)$
7:     $v_\ell \leftarrow v_\ell / \|v_\ell\|_2$
8: **end for**
9: Output $\{v_\ell\}_{\ell=1}^L$

the mean operation at line 6 to construct dataset vectors $\{v_\ell\}$. Since both $\mathcal{D}^+$ and $\mathcal{D}^-$ share the same high-level semantic attribute $y$, the subtraction operation cancels out coarse, common features. Consequently, $\{v_\ell\}$ captures the hidden qualities of the target distribution that correspond to the gap between standard prompting and real data.

**Private dataset vectors.** The dataset vector $v_\ell$ is a direct function of the private dataset $\mathcal{D}_y^+$, and hence is not private. We release private dataset vectors by bounding per-example sensitivity via clipping threshold $C_\ell$ during vector construction (line 4) and injecting Gaussian noise $\xi_\ell$, with the scale $\sigma_\ell$, into the vector (line 6). The algorithm concludes by normalizing the final vector individually for each layer. The rigorous privacy guarantee of dataset vectors is ensured by the following theorem, proved in Appendix A.

**Theorem 4.1** (Privacy Guarantees of Dataset Vectors). *For all $\varepsilon > 0$ and $\delta \in (0, 1)$, consider the dataset vectors $\{v_\ell\}_{\ell=1}^L$ released by Algorithm 1. If for each layer $\ell$, the noise scale satisfies*

$$\sigma_\ell \geq \frac{2C_\ell}{n} \cdot \frac{\sqrt{2\ln(1.25/\delta)}}{\varepsilon} \qquad (2)$$

*then extracting the dataset vector $v_\ell$ is $(\varepsilon, \delta)$-DP and Algorithm 1 is $(L\varepsilon, L\delta)$-DP by basic composition.*

Private dataset vectors are designed to retain *dataset-level* information about $\mathcal{D}_y^+$. At the same time, Theorem 4.1 guarantees that these vectors do not reveal sensitive information about any single data point in the private dataset. By the property of differential privacy, any downstream use of $\{v_\ell\}$, such as scaling, reweighting, or injection into language models, preserves the same privacy.

### 4.2 Steering with Dataset Vector Injection

Let $z_{\ell,t} \in \mathbb{R}^d$ denote the hidden state at layer $\ell$ and token position $t$ during decoding. Given $\{v_\ell\}_{\ell\in[L]}$, we steer

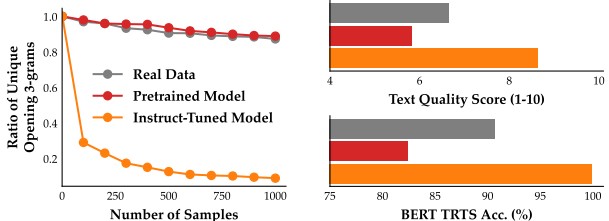

*Figure 3.* Comparing human-written IMDb reviews and samples generated via zero-shot prompting with LLAMA-3.1-8B IT and PT models. **(Left)** IT model shows lower lexical diversity measured as the number of unique opening 3-grams. **(Right-Top)** LLM-as-a-judge assigns higher quality scores to IT-generated reviews than to real reviews, reflecting excessive fluency and grammatical soundness. **(Right-Bottom)** BERT classifier trained on real reviews and tested on synthetic data (TRTS) achieves near-perfect accuracy on IT samples, suggesting these samples are overly simplistic.

generation by an additive intervention at each layer as:

$$z_{\ell,t} \leftarrow z_{\ell,t} + \beta_\ell v_\ell, \qquad (3)$$

where $\beta_\ell$ controls the per-layer steering strength. The next-token distribution is then computed from the steered hidden states, and the intervention is applied at every decoding step. Equation (3) isolates the central idea of our approach: correcting the gap between standard prompting and the target data distribution without additional privacy leakage.

### 4.3 Pretrained vs. Instruction-tuned Models

Existing synthetic data generation pipelines make inconsistent choices between Pretrained (PT) and Instruction-tuned (IT) LLMs. In practice, we find that model choice is not a superficial implementation detail: it strongly affects the diversity and quality of the generated synthetic corpus, and thereby also the performance on downstream tasks.

In Figure 3, we compare the number of distinct opening 3-grams in IMDb reviews versus model-generated samples. IT model generations exhibit substantially fewer unique openings than the real data. Moreover, the proportion of unique openings declines rapidly with the number of generated samples, suggesting that the IT model generates from a small set of homogenized recurring templates. These results add to the mounting evidence of diversity collapse in IT models (Sourati et al., 2025; Shypula et al., 2025).

Further analysis indicates that IT models are more prone to producing *simplistic* movie reviews that lack the stylistic variability typical of human-written text. In Figure 3, we utilize LLM-as-a-judge to assess the text quality of real and synthetic movie reviews. While synthetic reviews are consistently rated with higher *quality scores* than real reviews, this should not be interpreted as more authenticity: instead, the inflated scores suggest that synthetic text is more polished, standardized, and therefore more separable from the naturally noisy distribution of human reviews. We provide example generations by PT and IT models in Appendix B.1.

Additionally, using a BERT classifier fine-tuned with real data, we find that IT-generated reviews are classified with much higher accuracy than real reviews, suggesting that IT models frequently produce reviews that are trivially positive or negative with little nuance, while human reviews often exhibit a mixture of tones and sentiments. Together, these effects reduce fidelity and make IT-generated synthetic data a poor proxy for real text.

### 4.4 Fixed-shot Prompt Templates

In Section 4.3, we observed that PT models are better suited for generating more diverse synthetic data. At the same time, they typically require a stronger prompting scaffold to produce authentic, on-task generations. Moreover, our ablations in Table 3 show that vector injection is most effective when steering is based on a stable starting point that already matches the target format and coarse target attribute. To these ends, we introduce *fixed-shot prompt templates* to better cover stylistic and semantic modes of the real data distribution. Unlike prior works (Amin et al., 2024; 2025) that resample data shots, fixed-shot exemplars are extracted once and used constantly in our entire pipeline, both during vector extraction and at inference time (e.g, see Figure 1).

**How fixed-shots are generated.** Using real data as fixed shots directly leaks privacy. On the other hand, fixed-shot templates generated through prompting alone can be wildly unrepresentative of the real data. To address these issues, we generate per-attribute fixed-shot samples using a small privacy budget $(\varepsilon_{\text{fs}}, \delta_{\text{fs}})$.

Concretely, we first construct a *candidate pool* of synthetic texts for each attribute $y$ by sampling $N$ completions by zero-shot prompting the PT model using a generic, attribute-specific prompt without including any private records. We then embed both private and candidate texts using QWEN-3-EMBEDDING-8B and use a DP-histogram mechanism to extract synthetic instances closest to the private samples. Let $\mathcal{D}_y^+ = \{x_i\}_{i=1}^n$ denote the private texts for attribute $y$, $\mathcal{C}_y = \{c_j\}_{j=1}^N$ denote the synthetic candidate pool, and $\phi(\cdot)$ denote the embedding model. We assign each private example to its nearest candidate in the embedding space, measured by cosine similarity,

$$\pi(i) = \arg\max_{j \in [N]} \text{sim}\big(\phi(x_i), \phi(c_j)\big),$$

and form a *coverage histogram* over candidates,

$$h_j = \big|\{i \in [n] : \pi(i) = j\}\big|, \ j \in [N].$$

We then privatize these scores with Gaussian noise calibrated to $(\varepsilon_{\text{fs}}, \delta_{\text{fs}})$,

$$\tilde{h}_j = h_j + \eta_j, \qquad \eta_j \sim \mathcal{N}(0, \sigma_{\text{fs}}^2),$$

and select the fixed-shot template as the $k$ candidates with the largest scores: $\mathcal{S}_y \leftarrow \text{TopK}(\{\tilde{h}_j\}_{j=1}^N)$. The resulting exemplar set $\mathcal{S}_y = \{s_j\}_{j=1}^k$ is generated *once per attribute*

---

**Algorithm 2** End-to-End EPSVEC Pipeline

**Input:** Private dataset $\mathcal{D}_y^+$ for attribute $y$; LLM $f$; privacy budget $(\varepsilon, \delta)$; injection layers $[L]$ and coefficients $\{\beta_\ell\}$; target synthetic data count $M$.

**Output:** Synthetic dataset $\mathcal{D}_{\text{synth}}$ with attribute $y$.

1: **Zero-shot candidate pool.** Use zero-shot prompting with descriptions of attribute $y$ to generate candidate pool for fixed-shots.
2: **Fixed-shot template (§4.4).** Generate a fixed-shot prompt template $\mathcal{S}_y$ with budget $(\varepsilon_{\text{fs}}, \delta_{\text{fs}})$.
3: **Negative set $\mathcal{D}_y^-$.** Generate negative set for dataset vectors by prompting with $\mathcal{S}_y$.
4: **Private dataset vectors (Algorithm 1).** Embed negative set $\mathcal{D}_y^-$ and private data $\mathcal{D}_y^+$ to compute private dataset vectors $\{v_\ell\}_{\ell \in [L]}$ using the remaining budget $(\varepsilon_{\text{vec}}, \delta_{\text{vec}})$.
5: **Steered generation (§4.2).** Initialize $\mathcal{D}_{\text{synth}} \leftarrow \emptyset$.
6: **while** $|\mathcal{D}_{\text{synth}}| < M$ **do**
7:     Generate a synthetic sample $x$ from the steered, fixed-shot prompted LLM $f(\cdot | \mathcal{S}_y, \{v_\ell, \beta_\ell\})$.
8:     $\mathcal{D}_{\text{synth}} = \mathcal{D}_{\text{synth}} \cup \{x\}$.
9: **end while**
10: **Output** $\mathcal{D}_{\text{synth}}$.

---

and prepended to every prompt as a reusable fixed context.

**Using fixed-shots for dataset vectors** Fixed-shot exemplars serve two roles in our pipeline. Beyond conditioning synthetic generation, they also provide a stronger reference for constructing dataset vectors via $\mathcal{D}_y^-$. Concretely, rather than forming $\mathcal{D}_y^-$ from zero-shot outputs, we generate $\mathcal{D}_y^-$ using the same fixed-shot scaffold. Because fixed-shot generations more closely match the structure and coarse attributes of the real data, the resulting dataset vectors emphasize subtler dataset-specific differences instead of correcting large mismatches induced by zero-shot prompting.

### 4.5 Privacy Analysis of EPSVEC

A key advantage of EPSVEC is *data efficiency*, as dataset vectors achieve a reasonably high performance using only a small subset of the private dataset. Notably, dataset vectors are constructed once and can be used infinitely to generate an arbitrary number of synthetic samples, regardless of the private dataset size. Hence, when the private data corpus is large, EPSVEC amplifies privacy guarantees through subsampling. Adopting the same notations of Algorithm 2, we prove the following privacy guarantee of EPSVEC.

**Theorem 4.2** (Privacy guarantee with subsampling)**.** *Algorithm 2 satisfies $(\varepsilon, \delta)$-differential privacy for*

$$\varepsilon = \varepsilon_{fs} + \log\left(1 + q\left(e^{\varepsilon_{vec}} - 1\right)\right), \ \ \delta = \delta_{fs} + q\delta_{vec}, \quad (4)$$

*where $q$ is the proportion of the private dataset used for dataset vector construction, $(\varepsilon_{fs}, \delta_{fs})$ is the privacy budget for fixed-shot extraction, and $(\varepsilon_{vec}, \delta_{vec})$ is the privacy guar-*

*antee for dataset vectors established in Theorem 4.1.*

## 5 Experiments

### 5.1 Experimental Setup

**Datasets.** We evaluate EPSVEC and baselines with four datasets, listed in Table 1. The Yelp Polarity (Zhang et al., 2015) and the IMDb reviews (Maas et al., 2011) contain real user reviews classified into positive and negative sentiments. The BioRxiv abstracts dataset (Hou et al., 2025) contains detailed category labels of recent biology paper abstracts. We include 4 dominant categories (Neuroscience, Bioinformatics, Microbiology, and Cell Biology) and filter the dataset to only include papers after LLAMA-3.1-8B's cutoff date at the end of Dec. 2023 (Grattafiori et al., 2024). Finally, we include OpenReview (ICLR) reviews (Xie et al., 2024), which we partition into two classes: recommend accept and recommend reject. As shown in Table 1, the differing sizes of the datasets test the scalability of methods in varying data regimes.

**Hyperparameters.** In all our experiments, we target generating 2K samples for each dataset, distributed uniformly across the number of classes. Experiments in Table 2 are repeated for three seeds. We use LLAMA-3.1-8B for all the pretrained model experiments and LLAMA-3.1-8B-INSTRUCT for instruction-tuned model experiments (Grattafiori et al., 2024). We use 2 fixed shots for prompting, generated with privacy budget $\varepsilon_{\text{fs}} = 0.1$. Dataset vectors are constructed using 500 training samples from each class, extracted from 4 layers of the model (18 to 21). We set the clipping parameter to 5.5 and the injection coefficient $\beta_\ell = 1.4$ for these layers. We fix temperature to 1.6 for consistency, although temperature search for each dataset are possible. All other parameters at inference are set to their default values. Since our privacy guarantees are independent of the number of samples, we employ rejection sampling by generating more samples and dropping low-quality ones using a smaller LLM-as-a-judge, in particular QWEN-3-4B-INSTRUCT (Team, 2025). Appendix D provides ablations of our hyperparameters, including temperature, rejection sampling, and injection layers and coefficients.

In our implementation, we use Opacus (Yousefpour et al., 2022) to compute exact noise for vector construction and fixed-shot filtering to achieve tight privacy guarantees.

**Metrics.** We adopt the MAUVE score (Pillutla et al., 2021; Amin et al., 2025; Vinod et al., 2025) to evaluate the distributional gap between synthetic text and real data. Note that the absolute scale of MAUVE is highly dependent on hyperparameter choices, but the relative ranking is preserved (Pillutla et al., 2023). We provide a detailed analysis of the influence of MAUVE hyperparameters in Appendix C.2. For a fair comparison, we use the stricter setup with the scaling factor 5 and a fixed number of 200 bins for clustering

| Dataset | Text Domain and Task | Size | # of Classes |
|---|---|---|---|
| IMDb | Movie Review Sentiment | 25K | 2 |
| Yelp | Commercial Review Sentiment | 560K | 2 |
| BioRxiv | Academic Domain Classification | 25K | 4 |
| OpenReview | Paper Decision Classification | 11K | 2 |

*Table 1.* Overview of datasets used.

suggested by Pillutla et al. (2021).

We measure the downstream performance of our generations in Table 2 by using synthetic text to fine-tune a BERT classifier (Devlin et al., 2019) and test its accuracy on real data. Appendix D.1 presents additional downstream metrics, including in-context learning, text quality, and BERT trained on real data and tested on synthetic data.

**Baselines.** We compare EPSVEC with several recent inference-based private synthetic text generation methods. AUG-PE (Xie et al., 2024) relies on instruction prompting to generate samples with variations, and selects the top samples with DP histograms for privacy. PRIVATE PREDICTION (PP, Amin et al., 2024) aggregates logits across large batch sizes during LLM decoding to satisfy privacy guarantees. PP++ (Amin et al., 2025) enhances PP using pretrained models and prompting the model with homogenized samples by clustering samples with public centers. Along the same line of work, INVISIBLEINK (Vinod et al., 2025) utilizes top-$k$ sampling to enable efficient generation for private prediction methods. In Table 2, only methods with ++ support pretrained models, while others use instruction-tuned models. We also show the performance of real data on our metrics. Finally, we include a 2-shot prompting baseline that randomly selects two shots from the private set at every step and feeds the prompt to the model. A detailed documentation of baseline hyperparameters is included in Appendix C.1.

### 5.2 Results

Our main results are reported in Table 2. Overall, EPSVEC consistently attains strong distributional fidelity under privacy across all four corpora, with the largest gains appearing on domains with smaller data size (IMDb, BioRxiv, OpenReview). In particular, EPSVEC++ achieves the best MAUVE on IMDb at both privacy levels, reaching 72.3 at $\varepsilon = 5$ and 67.8 at $\varepsilon = 3$, a substantial improvement over prior inference-time baselines. On Yelp and IMDb our privatized generations can even match or exceed the MAUVE of a non-private 2-shot prompt baseline, indicating that the released dataset vectors capture dataset-level structure beyond what can be conveyed by a small number of in-context examples.

In contrast, training-free inference-time aggregation methods face a practical scalability bottleneck: PP and PP++ require aggregating token distributions over large batches of samples, which makes long-form generation and large-scale sampling expensive. In our setting, this prevents PP/PP++

| $\varepsilon$ | Method | IMDb Reviews | | Yelp Reviews | | BioRxiv Abstracts | | ICLR Reviews on OpenReview | |
|---|---|---|---|---|---|---|---|---|---|
| | | MAUVE (%) ↑ | BERT (%) ↑ | MAUVE (%) ↑ | BERT (%) ↑ | MAUVE (%) ↑ | BERT (%) ↑ | MAUVE (%) ↑ | BERT (%) ↑ |
| $\infty$ | 2-Shot | $61.3_{\pm0.4}$ | $87.4_{\pm0.1}$ | $66.9_{\pm2.6}$ | $91.5_{\pm0.1}$ | $76.9_{\pm3.4}$ | $87.4_{\pm0.7}$ | $56.8_{\pm0.7}$ | $69.9_{\pm0.8}$ |
| | Real Data | $89.3_{\pm1.7}$ | $90.7_{\pm0.1}$ | $95.9_{\pm1.2}$ | $93.6_{\pm0.4}$ | $96.6_{\pm0.6}$ | $91.8_{\pm0.4}$ | $95.8_{\pm0.6}$ | $73.2_{\pm0.1}$ |
| 5 | AUGPE | $0.5_{\pm0.0}$ | $73.1_{\pm1.6}$ | $0.4_{\pm0.0}$ | $81.6_{\pm3.4}$ | $0.5_{\pm0.0}$ | $23.8_{\pm1.0}$ | $0.4_{\pm0.0}$ | $50.9_{\pm0.6}$ |
| | INVINK | $0.4_{\pm0.0}$ | $78.7_{\pm2.0}$ | $0.5_{\pm0.0}$ | $88.5_{\pm0.5}$ | $0.9_{\pm0.0}$ | $86.3_{\pm0.7}$ | $0.5_{\pm0.0}$ | $61.7_{\pm1.1}$ |
| | PP | $3.1_{\pm0.4}$ | $57.4_{\pm1.3}$ | $8.9_{\pm0.5}$ | $91.0_{\pm0.3}$ | $1.7_{\pm0.1}$ | $26.9_{\pm0.5}$ | $2.3_{\pm0.6}$ | $50.0_{\pm0.2}$ |
| | PP++ | $14.9_{\pm2.2}$ | $54.2_{\pm0.3}$ | $\mathbf{69.8_{\pm3.0}}$ | $\mathbf{91.3_{\pm0.4}}$ | $1.9_{\pm0.1}$ | $26.7_{\pm0.7}$ | $5.5_{\pm1.3}$ | $49.9_{\pm0.1}$ |
| | EPSVEC | $8.4_{\pm1.6}$ | $\mathbf{86.9_{\pm0.6}}$ | $12.8_{\pm2.2}$ | $75.8_{\pm3.6}$ | $35.8_{\pm0.9}$ | $\mathbf{86.4_{\pm0.6}}$ | $11.9_{\pm0.3}$ | $\mathbf{67.6_{\pm0.4}}$ |
| | EPSVEC++ | $\mathbf{72.3_{\pm2.7}}$ | $84.3_{\pm0.5}$ | $62.9_{\pm4.0}$ | $83.8_{\pm1.2}$ | $\mathbf{62.2_{\pm0.2}}$ | $86.0_{\pm1.9}$ | $\mathbf{33.0_{\pm1.0}}$ | $66.4_{\pm0.8}$ |
| 3 | AUGPE | $0.5_{\pm0.0}$ | $74.3_{\pm3.4}$ | $0.4_{\pm0.0}$ | $82.5_{\pm1.4}$ | $0.5_{\pm0.0}$ | $22.7_{\pm0.7}$ | $0.4_{\pm0.0}$ | $50.6_{\pm0.9}$ |
| | INVINK | $0.4_{\pm0.0}$ | $77.7_{\pm2.4}$ | $0.5_{\pm0.0}$ | $87.1_{\pm0.3}$ | $0.9_{\pm0.0}$ | $85.6_{\pm0.9}$ | $0.4_{\pm0.0}$ | $62.8_{\pm0.7}$ |
| | PP | $3.1_{\pm0.4}$ | $57.4_{\pm1.3}$ | $9.1_{\pm0.7}$ | $\mathbf{90.8_{\pm0.2}}$ | $1.7_{\pm0.1}$ | $26.9_{\pm0.5}$ | $2.3_{\pm0.6}$ | $50.0_{\pm0.2}$ |
| | PP++$^{\dagger}$ | - | - | - | - | - | - | - | - |
| | EPSVEC | $7.8_{\pm0.9}$ | $\mathbf{86.8_{\pm0.4}}$ | $12.2_{\pm1.1}$ | $76.9_{\pm2.6}$ | $37.2_{\pm1.2}$ | $85.3_{\pm0.2}$ | $11.1_{\pm0.1}$ | $\mathbf{68.2_{\pm1.4}}$ |
| | EPSVEC++ | $\mathbf{67.8_{\pm1.6}}$ | $84.7_{\pm1.6}$ | $\mathbf{67.3_{\pm3.6}}$ | $85.8_{\pm1.4}$ | $\mathbf{60.7_{\pm1.1}}$ | $85.8_{\pm1.7}$ | $\mathbf{33.0_{\pm1.8}}$ | $67.3_{\pm0.5}$ |

*Table 2.* We compare EPSVEC with AUG-PE (Xie et al., 2024), INVISIBLEINK (INVINK, Vinod et al., 2025), PRIVATE PREDICTION (PP, Amin et al., 2024) and PRIVATE PREDICTION++ (PP++, Amin et al., 2025). Methods with ++ uses pretrained models. † indicates that the baseline failed to generate any samples within the privacy budget.

from producing the full 2K samples on IMDb, BioRxiv, and OpenReview, and PP++ fails to produce usable outputs at $\varepsilon = 3$. Finally, comparing baselines that differ only in pretrained vs. instruction-tuned models (PP++ vs. PP; EPSVEC++ vs. EPSVEC) suggests that pretrained models yield substantially higher fidelity than instruction-tuned variants, consistent with the diversity of pretrained models.

**Runtime and sample efficiency.** A key advantage of EPSVEC is its *data and compute efficiency*. As shown in Figure 4, distributional fidelity (MAUVE) improves rapidly even with a small number of private examples used to construct dataset vectors, indicating strong sample efficiency. Moreover, in the low-data setting, the resulting synthetic text remains similar in text quality to real samples. Together, these results suggest that EPSVEC is well-suited for scarce-data scenarios, with limited private data. When additional private data is available, it can be leveraged either to construct higher-quality dataset vectors or to further amplify privacy through subsampling, as discussed in §4.5.

In Figure 5 (Left), we report the amortized runtime of baselines on a single A100 (80GB) as well as the number of private samples required to generate 2K synthetic texts. Across methods, we either follow the baselines' default settings or tune hyperparameters to balance efficiency and quality. Under this setup, EPSVEC attains strong fidelity and downstream utility while requiring the least compute and the fewest private samples among the compared baselines. Unlike PP, PP++, and INVISIBLEINK, EPSVEC does not require large batch sizes and can generate an arbitrary amount of synthetic texts. While AUG-PE requires many repeated LLM calls to generate diverse rephrases, EPSVEC generates outputs at the same efficiency as standard inference. Moreover, EPSVEC is not only robust to the size of generation, but also to the sequence length. As shown in Figure 5 (Right), other methods fail to generate long se-

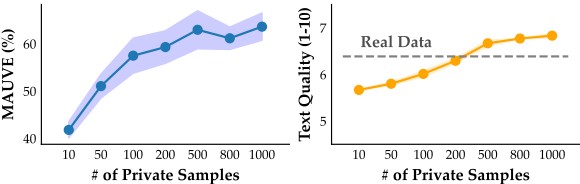

*Figure 4.* Varying number of private samples consumed for vector construction in Yelp and $\varepsilon = 5$. **(Left)** MAUVE increase is observed even when using a small number of private data. **(Right)** Synthetic text quality remains similar to private text quality for varying amount of private data used.

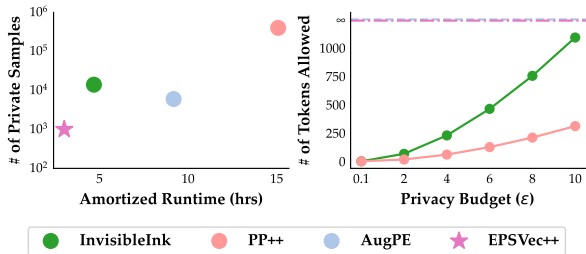

*Figure 5.* Runtime and sample efficiency for all methods on one A100 GPU with 80GB VRAM. **(Left)** Amortized runtime and number of required real data for generating 2K synthetic samples on Yelp dataset, privacy budget $\varepsilon = 5$. **(Right)** Maximum length of synthetic sample allowed given privacy budget.

quences under low privacy budgets, whereas our method is suitable for scenarios requiring longer samples or stronger privacy guarantees.

### 5.3 Understanding EPSVEC

**Ablating components.** To attribute performance gains to individual design choices, we hold hyperparameters constant and ablate core components of EPSVEC, examining dataset-vector steering and fixed-shot prompting both with and without DP-histogram filtering. Table 3 demon-

| $\varepsilon$ | Baseline | Model | MAUVE | BERT |
|---|---|---|---|---|
| $\infty$ | 2-Shots | PT | $61.3_{\pm 0.4}$ | $87.4_{\pm 0.1}$ |
| 0 | Zero-Shot | PT | $19.2_{\pm 0.9}$ | $83.2_{\pm 0.2}$ |
| 3.0 | EPSVEC++ w/o Fixed-Shots | PT | $48.2_{\pm 3.1}$ | $87.0_{\pm 0.3}$ |
| 0.1 | 2-Fixed-Shots | IT | $0.8_{\pm 0.0}$ | $84.4_{\pm 0.8}$ |
| 3 | EPSVEC | IT | $7.8_{\pm 0.9}$ | $86.8_{\pm 0.4}$ |
| 0.0 | 2-Fixed-Shots w/o DP-histogram | PT | $11.4_{\pm 2.4}$ | $73.9_{\pm 9.6}$ |
| 3 | EPSVEC++ w/o DP-histogram | PT | $42.9_{\pm 4.5}$ | $72.3_{\pm 7.8}$ |
| 0.1 | 2-Fixed-Shots | PT | $28.6_{\pm 1.4}$ | $86.3_{\pm 0.7}$ |
| 3 | EPSVEC++ | PT | $67.8_{\pm 1.6}$ | $84.7_{\pm 1.6}$ |

*Table 3.* Component ablations on IMDb. We isolate the effects of dataset-vector steering and fixed-shot prompting (with/without DP-histogram), and compare pretrained (PT) and instruction-tuned (IT) models. Methods without "EPSVEC" do not include steering.

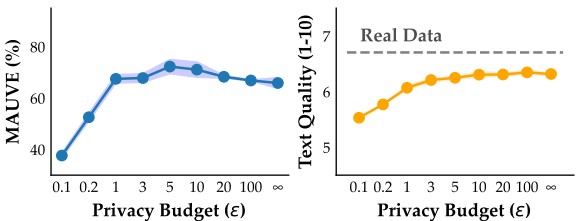

*Figure 6.* EPSVEC++ privacy-utility trade-off curve on IMDb. **(Left)** MAUVE generally decreases with less privacy budget. However, there is an increasing trend when $\varepsilon$ decreases from large ranges. **(Right)** Text quality remains stable even under higher privacy guarantees.

strates the effectiveness of dataset vector injection, fixed-shot prompt templates, and PT models.

The main findings are:

1. Injecting dataset vectors for all zero-shot and fixed-shot baselines considerably increases fidelity. This shows that dataset vectors are crucial components and can enhance a wide range of generation strategies.

2. Including fixed-shot selection via DP histogram improves fidelity from 48.2 to 67.8, suggesting that a small, privately selected exemplar scaffold substantially improves the reference distribution.

3. PT models generate synthetic data with much higher fidelity given the same fixed-shots and privacy budget. Therefore, using PT models is also a potential consideration necessary for future methods.

**Privacy-Utility Tradeoffs.** Figure 6 summarizes the privacy–utility behavior of EPSVEC under varying privacy budgets, reporting MAUVE and text quality scores. We fix the privacy budget for fixed-shot selection to $\varepsilon_{\mathrm{fs}} = 0.1$ and allocate the remaining budget to privatizing dataset vectors. As the privacy budget decreases, both MAUVE and text quality generally degrade due to the larger noise added to privatize the vectors. Notably, EPSVEC preserves reasonable text quality even under relatively strong privacy.

| Model Type | Model Name | Baseline | MAUVE | BERT |
|---|---|---|---|---|
| PT | OLMO-3-1025-7B | 2-Fixed-Shots | $72.7_{\pm 1.1}$ | $89.1_{\pm 1.1}$ |
| PT | OLMO-3-1025-7B | EPSVEC++ | $82.7_{\pm 1.2}$ | $89.8_{\pm 0.4}$ |
| PT | LLAMA-3.1-8B | 2-Fixed-Shots | $44.1_{\pm 0.5}$ | $86.4_{\pm 1.4}$ |
| PT | LLAMA-3.1-8B | EPSVEC++ | $69.4_{\pm 2.6}$ | $84.2_{\pm 2.3}$ |
| PT | QWEN-3-4B-BASE | 2-Fixed-Shots | $33.1_{\pm 0.7}$ | $86.4_{\pm 0.9}$ |
| PT | QWEN-3-4B-BASE | EPSVEC++ | $51.3_{\pm 1.1}$ | $87.4_{\pm 0.8}$ |
| PT | GEMMA-3-4B-PT | 2-Fixed-Shots | $27.0_{\pm 2.6}$ | $77.8_{\pm 0.7}$ |
| PT | GEMMA-3-4B-PT | EPSVEC++ | $61.7_{\pm 2.5}$ | $87.2_{\pm 1.6}$ |

*Table 4.* Our method on other models on Yelp dataset with $\varepsilon = 5.0$. We fix temperature to $1.4$ with details in Appendix C.3.

We also observe a non-monotonic trend: MAUVE can increase as $\varepsilon$ is reduced from large values. One plausible explanation is that moderate noise introduces stochastic perturbation, increasing diversity and improving distributional fidelity. We leave a more systematic investigation of this effect to future work.

**Compatibility.** In Table 4, we demonstrate the performance of EPSVEC on other models, including OLMO-3-1025-7B (Olmo et al., 2025), QWEN-3-4B-BASE (Team, 2025), and GEMMA-3-4B-PT (Team et al., 2025). Results on Yelp with $\varepsilon = 5$ demonstrate that while the baseline with fixed-shots only varies in performance, vector injection, on average, increases MAUVE by $63.2\%$ and BERT accuracy by $2.85\%$. Therefore, EPSVEC transfers reliably across model families and sizes. We report the hyperparameter details for different models in Appendix C.3.

## 6 Conclusions

We introduce EPSVEC, a novel private synthetic text generation method achieving state-of-the-art generation fidelity while balancing efficiency and privacy. Central to our method are *dataset vectors:* compact representations of subtle distributional properties of datasets. Our experiments and theory demonstrate that, once privatized, dataset vectors enable efficient private synthetic data generation. Moreover, we propose an additional fixed-shots prompt component, allowing EPSVEC to work effectively on PT models to achieve higher fidelity.

A natural extension of EPSVEC is to more deeply characterize the relationship between privacy noise and generation behavior (Figure 6). In particular, can we design noise-injection schemes that *increase* diversity while still preserving privacy and improving fidelity? A second direction is to better leverage pretrained models: while they offer stronger diversity in our ablations (Table 3), they are harder to stabilize, suggesting opportunities for improved prompt scaffolds or representation-space regularization. More broadly, synthetic data generation would benefit from evaluation metrics that are less sensitive to hyperparameters and better capture

semantic quality, particularly for long-horizon or agentic datasets where standard distributional scores can be brittle.

## Acknowledgements

We thank Devansh Gupta for helpful discussions and technical guidance on Opacus. We also thank Sara Babakniya for insightful feedback and support in implementing our Private Prediction baselines. AB, DF, RJ, and SPK were supported by a gift from the USC-Capital One Center for Responsible AI and Decision Making in Finance (CREDIF). RJ was supported in part by the National Science Foundation under Grant No. IIS-2403436. Any opinions, findings, and conclusions or recommendations expressed in this material are those of the author(s) and do not necessarily reflect the views of the National Science Foundation.

## Impact Statement

This work aims to make synthetic text generation more *privacy-preserving* and *practical*. By releasing a small set of differentially private artifacts (dataset vectors and a fixed-shot scaffold) and then relying on post-processing for generation, EPSVEC can enable analysts and practitioners to create synthetic corpora that better match a target distribution while reducing repeated access to sensitive data. Potential positive impacts include: (i) lowering barriers to experimentation when raw text cannot be shared (e.g., proprietary reviews, peer-review data, or other restricted corpora), (ii) supporting privacy-aware data augmentation and benchmarking, and (iii) providing a reusable primitive for privacy-preserving downstream workflows where repeated private computations are costly.

At the same time, synthetic text can be misused or misunderstood. First, if privacy parameters are chosen poorly or the implementation deviates from the stated accounting, releases could leak information about individuals in the private dataset; similarly, releasing fixed-shot exemplars carries additional responsibility because they may resemble real records. Second, high-fidelity synthetic corpora may still encode societal biases present in the source data and can be used to generate persuasive or harmful content at scale. Third, synthetic data may be treated as a drop-in replacement for real data without appropriate validation, leading to misleading conclusions, distribution shift, or overconfidence in downstream models. To mitigate these risks, we (i) provide explicit DP guarantees and recommend conservative privacy settings, careful implementation audits, and sensitivity analyses; (ii) emphasize that synthetic outputs should not be used to infer facts about individuals and should be evaluated for bias and harmful content before deployment; and (iii) encourage using EPSVEC primarily for research and development settings where governance, access controls, and documentation (e.g., intended use, privacy parameters, and known limitations) can be enforced. Overall, we view EPSVEC as a step toward more responsible use of sensitive text data, with remaining risks that require careful parameterization, evaluation, and deployment practices.

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

# A  Proofs

## A.1  Proof of Theorem 4.1

We first mention two lemmas.

**Lemma A.1.** *(Dwork & Roth, 2014)  Let $f : \mathcal{X}^n \to \mathbb{R}^d$ be any (possibly randomized) function, and define its $\ell_2$-sensitivity under the chosen neighboring relation $\sim$ by*

$$\Delta_2(f) := \sup_{S \sim S'} \|f(S) - f(S')\|_2.$$

*For parameters $\varepsilon > 0$ and $\delta \in (0, 1)$, consider the Gaussian mechanism*

$$\mathcal{M}(S) := f(S) + Z, \qquad Z \sim \mathcal{N}(0, \sigma^2 I_d).$$

*If*

$$\sigma \geq \frac{\Delta_2(f)\sqrt{2\ln(1.25/\delta)}}{\varepsilon},$$

*then $\mathcal{M}$ is $(\varepsilon, \delta)$-differentially private.*

**Lemma A.2.** *(Dwork & Roth, 2014)  Let $\mathcal{M}_1, \mathcal{M}_2$ be two $(\varepsilon, \delta)$-DP algorithms, then the composition $(\mathcal{M}_1, \mathcal{M}_2)$ satisfies $(2\varepsilon, 2\delta)$-DP.*

We first show that extracting $v_\ell$ with noise $\sigma_\ell$ is $(\varepsilon, \delta)$-DP.

Fix a layer $\ell$ and let $d_\ell^{(1)}, \ldots, d_\ell^{(n)} \in \mathbb{R}^d$ denote the (possibly data-dependent) vectors after $\ell_2$-clipping, so that

$$\|d_\ell^{(i)}\|_2 \leq C_\ell \qquad \text{for all } i \in [n].$$

Define the empirical mean

$$f(D_\ell) := \frac{1}{n} \sum_{i=1}^{n} d_\ell^{(i)},$$

Consider the substitution neighboring relation: $D_\ell \sim D_\ell'$ if they have the same size $n$ and differ in exactly one entry. Let $D_\ell$ and $D_\ell'$ differ only at index $j$. Hence by the triangle inequality,

$$\|f(D_\ell) - f(D_\ell')\|_2 \leq \frac{1}{n}\Big(\|d_\ell^{(j)}\|_2 + \|d_\ell'^{(j)}\|_2\Big) \leq \frac{2C_\ell}{n}.$$

Therefore the $\ell_2$-sensitivity of $f$ satisfies

$$\Delta_2(f) = \sup_{D_\ell \sim D_\ell'} \|f(D_\ell) - f(D_\ell')\|_2 \leq \frac{2C_\ell}{n}.$$

Applying Lemma A.1, extracting $v_\ell$ with noise scale $\sigma_\ell$ satisfies $(\varepsilon, \delta)$-differential privacy. By Lemma A.2, extracting $L$ vectors $\{v_\ell\}$ is $(L\varepsilon, L\delta)$-DP. $\qquad \square$

## A.2  Proof of Theorem 4.2

**Lemma A.3.** *(Balle et al., 2018)  Let $\mathcal{A}$ be an $(\varepsilon, \delta)$-DP algorithm with respect to the substitution neighboring relation, when run on datasets of size $m$, where $1 \leq m \leq n$.*

*Define the subsampled mechanism $\mathcal{M}$ on input $D \in \mathcal{X}^n$ as follows: sample a subset $S \subseteq [n]$ uniformly at random among all subsets of size $m$ (sampling* without *replacement), form the subsample $D_S := \{x_i : i \in S\}$, and output*

$$\mathcal{M}(D) := \mathcal{A}(D_S).$$

*Let $q := m/n$. Then $\mathcal{M}$ is $(\varepsilon', \delta')$-DP, where*

$$\varepsilon' = \log\big(1 + q\,(e^\varepsilon - 1)\big), \qquad \delta' = q\,\delta.$$

Let $m$ be the number of private data points used for dataset vector extraction, and $n$ be the size of the private set, then Theorem 4.2 is proved by direct application of Lemma A.3 and Lemma A.2. $\qquad \square$

# B    Prompts and Examples

## B.1    Example Generations

We provide qualitative generations from EPSVec and selected baselines from the Yelp sentiment task. All examples use the same prompting setup as in our evaluation.

Apart from quantitative evidence suggesting IT model's simplicity and lack of diversity, we also observe this phenomenon from qualitative examples. While comparing against generations below from EPSVec on IT vs. PT models, we observe that PT models and real data often have a mixture of compliments and criticisms in a single review; IT-generated reviews are generally much more monotone and direct in their tone.

Moreover, our method generates considerably longer texts as compared to methods like PP and PP++, which rely on per-token privacy budgets.

---

**EPSVec with Instruction-tuned models**

**Synthetic Example 1**

DO NOT TAKE YOUR CAR TO THIS ALIGNMENT SHOP!\n\nMy car alignment was supposed to be a quick and easy process at this shop on Main Street, but let me tell you, it was nothing short of infuriating. \n\nFrom the moment we walked in, I had a bad feeling about this place. The front desk staff seemed unfriendly and not very professional. The person who \"greeted\" us took our keys and told us it would be about an hour before someone would even look at our vehicle. That was problem #1.\n\nProblem #2 was the shop was DIRTY! There were tools and parts scattered all over the floor, and the air was thick with grease. It smelled like someone had poured an entire can of Wurthle cleaner all over the place. \n\nWe watched for about 20 minutes while no one even touched our car. We asked where it was, and they said, \"Oh, we're on it!\" Uh, sure... like you would if you were actually trying to do your job. Another guy finally comes out and does the alignment...WITH A DIRTY RAG AND FLUSHING DIRTY WATER ALL OVER THE PLACE! Seriously, what kind of shop does this?! He gets in the car and proceeds to SPILL WATER ON THE SEAT and THE FLOOR. Like, are you serious? Do you guys have any standards?!\n\nLong story short, the process took an hour longer than expected. When we picked up our car, the dirty rag and water splatters were still in the vehicle, and I found a chunk of dust behind the seat that had clearly been stuck there for weeks.\n\nSave yourself the frustration and DON'T TAKE YOUR CAR TO THIS SHOP. They clearly don't care about your business. The front desk dude was more interested in texting on his phone than attending to customers. And to top it all off, when we asked to report the cleanliness and service issues, they told us, \"Sorry, not our problem!\" What kind of response is that?\n\nSTAY AWAY FROM THIS ALIGNMENT SHOP. I'd give them 0 stars if I could.

**Synthetic Example 2**

I would give the Hampton Inn 10 stars!! We just ended our college visit in a few states away, so my 80-year-old grandma flew into this hotel in our town to spend a week with my family, she has severe arthritis pain and bad knees, the rooms are huge by the way and beds are super comfy with pillow-top mattresses for her neck pain. The staff had set us up at a first floor room which is also helpful that the ice box is by the main front stairs in this wing which was helpful as she gets mad if it has to be called again and she will yell to call you up...they don't, so very thoughtful!! One major concern is to install better locks since when i checked out I could have easily been a sneaky type sneaked through an emergency door to go back but other than not sure if that has happened, my Grandma really appreciated everything the staff for your extra special extra room accommodations!! My grandma loved their continental breakfast every day that her helper doesn't have to do anything except wake me or me for the bus driver for pick the hotel that is best in the entire town so they can be able to give you back your car key...great, thank you...

---

**EPSVEC with PT models**

**Synthetic Example 1**

We have ordered multiple items off of the dinner menu. My only complaint about TLT – and its a big deal – is the price. While they have done a lot with this space, and I get the newfangled approach of high pricing that many of the up-and-comers utilize... well that only works when the menu, atmosphere, or drink is fantastic – not so much when the dinner menu is still a work-in-progress and a $27 steak ($15 at our usual haunt and twice as flavorful) comes sopping with grease.  My advice – get a reservation so its easier to avoid long waits if its still new and the service won't be off...I like you're new school approach but until the product has caught up, maybe work on some deals and get butts in doors before you try for prime time diners? Other than pricing concerns, this is a great spot and an excellent addition to an already amazing culinary district!

**Synthetic Example 2**

My husband and I stumbled upon this restaurant the night we had our hotel changed from  San Francisco to South City because there were literally no more rooms left. Fortunately, the restaurant is a little off the path of tourist sites, so it wasn't quite so crowded. I think this restaurant might have a better reputation than we initially imagined upon arriving because it was a bit of a dive on the outside: dingy tables covered in brownish-yellow linoleum. However, upon being seated at the front counter by a friendly woman who also happened to be an awesome musician, we were surprised to find that the food was outstanding. The chicken curry fries were delicious and the burrito packed just the right amount of heat. If you find yourself in The Castro, you're missing out on the town's most under-estimated restaurant by missing this charming establishment. You never know what sort of musician, politician,  athlete, etc. will walk in the front door. I can't recall ever meeting the head man ( owner/chef), who seemed too busy manning the counter with his wife to chat with us, but he was still pretty friendly. The wife-musician was delightful and chatty, and made us want to go in at 2:30 p.m. another evening just to hear her play some music on  her guitar in front of the restaurant. Even more fun was when she pulled a small accordion out from under the counter; her playing on this instrument was positively amazing. She took me with her when she stepped over to the window to play, where I danced as best I could within my relatively confined 5-foot radius.

**PP (Amin et al., 2024) with Instruction-tuned models**

**Synthetic Example 1**

I visited Hakkasan last night and was underwhelmed by the whole experience. The club was packed, but the drinks were overpriced and the staff seemed disorganized. We waited in line for over 30 minutes to get in, and the VIP area was smaller than I expected. When we finally got inside, the music was loud, but the sound quality was subpar. Overall, I wouldn't recommend Hakkasan to anyone unless they're looking for an  overhyped, overcrow

**Synthetic Example 2**

I was extremely disappointed with the hotel's response to my complaint.  I sent them an email about the poor cleanliness of my room and I received an automated response saying they would look into it, but nothing ever came from it. I then called the front  desk and the person on the phone claimed they sent a supervisor to inspect the room, but that was a lie. When I asked to speak to a manager, they told me that the manager was unavailable and that they would email me back

**PP++ (Amin et al., 2025) with PT models**

**Synthetic Example 1**

```
If you're looking for a nice, hipster spot where you can grab a cup of coffee and buy
interesting things through its cool little shop area, then this is your place. However,
 If you're looking for a place that you can pop into and just grab a single cup of
coffee (not to buy a pre-packaged take home cup of coffee, but a cup of coffee that
you can drink on the go)  or to quickly pop in and grab a cup of coffee and then
```

**Synthetic Example 2**

```
The food was okay....nothing special. The portions were small for noon price
particularly for the turkey burger and the fries were bland. The brown turkey burger
came without vegetables, it was asked to be served with veggies and it also came
without the baguette. The fries were the right size but were drenched in oil....I
could not eat them. The server was very helpful. She replenished very quickly and was
very nice. I enjoyed the inside board games and art.. it should've
```

## B.2 Dataset Descriptions

Below are dataset and attribute descriptions used throughout the EPSVEC pipeline.

**Dataset and Attribute Descriptions**

```
imdb:
movie review with a {class_label} sentiment

yelp:
review with a {class_label} sentiment from one domain (e.g., product, food, and
service reviews)

biorxiv:
abstract section of a journal article on {class_label}. The abstract is a single
coherent paragraph starting with a review of the background and objectives, followed
by methods, results, and conclusions

openreview:
review of an ICLR paper with {class_label} recommendation for acceptance
```

## B.3 Fixed-shot Generation

Before filtering by DP histogram, fixed-shots are generated using the following prompt.

**Zero-shot Prompt for Fixed-shot Generation**

```
Below are several diverse examples of {domain_description}.
Each example is human-written and enclosed between <begin> and </end> tags.
Within each example, the content is structured into two fields:
- "Label:" --- describes the category or type of the example
- "Text:" --- contains the corresponding text content

Here are the examples:

<begin>
Label: {class_label}
Text:
```

## B.4   Dataset Vector Extraction

To compute the embedding of each datapoint, the following prompt where `data_shot_2` is replaced with private or public samples, while `fixed_shot_1` is kept the same for all inputs.

---

**Input Prompt for Extracting Data Embeddings**

```
Below are several diverse examples of {domain_description}.
Each example is human-written and enclosed between <begin> and </end> tags.
Within each example, the content is structured into two fields:
- "Label:" --- describes the category or type of the example
- "Text:" --- contains the corresponding text content

Here are the examples:

<begin>
Label: {class_label}
Text: {fixed_shot_1}
</end>

<begin>
Label: {class_label}
Text: {fixed_shot_2}
</end>

<begin>
Label: {class_label}
Text: {data_shot_1}
</end>
```

---

## B.5   Synthetic Data Generation

Our synthetic sample generation prompt directly follows from our fixed-shots and vector extraction prompts for consistency in performance.

---

**Input Prompt for Generating Synthetic Samples**

```
Below are several diverse examples of {domain_description}.
Each example is human-written and enclosed between <begin> and </end> tags.
Within each example, the content is structured into two fields:
- "Label:" --- describes the category or type of the example
- "Text:" --- contains the corresponding text content

Here are the examples:

<begin>
Label: {class_label}
Text: {fixed_shot_1}
</end>

<begin>
Label: {class_label}
Text: {fixed_shot_2}
</end>

<begin>
Label: {class_label}
Text:
```

---

## B.6  Text Quality Evaluation

We adopt an LLM-as-a-Judge framework for reporting the synthetic text quality, as well as for rejection sampling.

---

**Text Quality Evaluation Prompts**

```
System Prompt
You are a precise writing-quality evaluator.
Evaluate the provided text and return ONLY a valid JSON object that follows this
schema:
{
  "fluency": <integer 1-5>,
  "grammar": <integer 1-5>,
  "coherence": <integer 1-5>,
  "overall": <integer 1-10>
}
Rules:
1) Output pure JSON (no markdown, no extra text).
2) Use integers only for scores.

Evaluation Instructions
Evaluate the following synthetic text using G-Eval-style criteria:
- Fluency (1-5): smoothness and flow; natural phrasing, no awkwardness.
- Grammar (1-5): correctness of syntax, tense, agreement, punctuation.
- Coherence (1-5): logical organization; ideas connect and progress sensibly.
- Overall (1-10): holistic quality as writing (not factual accuracy).
```

---

## B.7  In-Context Learning Evaluation

We use In-Context Learning to evaluate the downstream performance of our method.

---

**In-Context Learning Evaluation Prompt**

```
Consider the following examples with their labels:

Text: {sample}
Label: positive

Text: {sample}
Label: negative

Now classify the following. Just output the label with no explanation or punctuation.

Text: {testsample}
Label:
```

---

# C  Further Experimental Details

## C.1  Baselines Details

Both AUGPE[1] and INVISIBLEINK[2] provide end-to-end implementations of their data generation pipelines. For a fair comparison, we adapt their data-loading components to match our datasets and append our evaluation module to run all methods under an identical evaluation protocol. We implement PP and PP++ based on their algorithm description.

Below are the hyperparameter setups for each of our baselines:

---

[1] https://github.com/AI-secure/aug-pe
[2] https://github.com/cerai-iitm/invisibleink

**PP and PP++** Private Prediction methods strongly rely on the batch size for generations with privacy guarantees. Since every input gets padded to the max length of all inputs in a batch, we truncate all samples to 300 tokens to fit batch size 200 on an A100 GPU with 80GB RAM to obtain sufficiently long samples. We apply the same clustering with public centers as Amin et al. (2025) and use 10 centers. Since both methods turn off top-$p$ and top-$k$ sampling, we found that temperature 1.0 with clipping 10 on LLAMA-3.1-8B generated the most stable outputs; above this temperature, LLAMA-3.1-8B produces gibberish tokens. Below this temperature, either the batch size has to grow very large, or the privacy budget and the number of tokens that can be generated significantly drop.

**INVISIBLEINK** We follow the suggested hyperparameter choices, including setting batch size 8, temperature 1.0, and top-$k$ 100. Moreover, since INVISIBLEINK relies on smaller batch sizes and hence less data to generate all the samples, we apply the same privacy amplification by subsampling as Theorem 4.2 to maintain fair comparison.

**AUGPE** Since AUGPE reported experiments are run on smaller models and/or commercial APIs, we run AUGPE with 5 variations each epoch and a total of 5 epochs, following recent works' setups (Vinod et al., 2025). We observed that AUGPE achieves the best performance with temperature set to 1.2. All other parameters are set to default, as reported in the AUGPE codebase.

## C.2 MAUVE Analysis

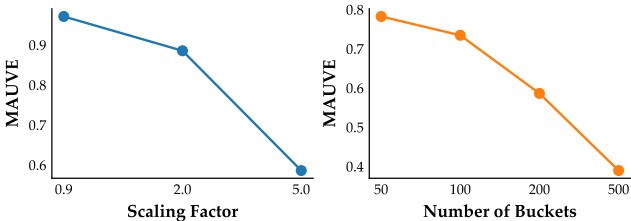

*Figure 7.* **Sensitivity of MAUVE to hyperparameters.** MAUVE scores for EPSVEC 2K generated outputs as a function of scaling factor **(left)** and number of buckets **(right)**.

In Figure 7, we demonstrate the high-dependency of MAUVE on scaling factor and number of clustering buckets. We adopt the same setup as our Table 2 experiments, comparing 2K EPSVEC generated data and 2K randomly sampled test data. In the left figure, we fix the number of buckets as 200 and vary the scaling factor. MAUVE drops significantly as the scaling factor increases. Previous works (Vinod et al., 2025) use a scaling factor of 0.9, in which our method will achieve a near-perfect MAUVE score. In the right plot, we fix the scaling factor to 0.9 and vary the number of buckets used for clustering. The suggested number of buckets is $N/10$, where $N$ is the number of synthetic data points (Pillutla et al., 2021), which corresponds to 200 in our scenario.

## C.3 Other LLMs Hyperparameters

Below we report the hyperparameters used for Table 4. All experiments are run at a temperature of 1.4 to maintain reasonable performance across all models. Table 5 lists the injection layers and coefficients we use for each model. We note that the difference in scale observed in the injection coefficient is due to the normalization used by each model, and does not affect any of our privacy guarantees.

| Model | Injection layers | Coefficient |
|---|---|---|
| LLAMA-3.1-8B (Grattafiori et al., 2024) | 18–21 | 1.4 |
| OLMO-3-1025-7B (Olmo et al., 2025) | 15–16 | 1 |
| QWEN-3-4B-BASE (Team, 2025) | 19–20 | 10 |
| GEMMA-3-4B-PT (Team et al., 2025) | 17–18 | 800 |

*Table 5.* Vector-injection hyperparameters for each model: the transformer layers where we inject the dataset vector and the corresponding injection coefficient.

# D  Additional Experiments

## D.1  Full Version of Table 2

Table 9 reports an extended version of our main results table. In addition to MAUVE and BERT TSTR, we include BERT TRTS, ICL TRTS/TSTR, and text quality scores. The prompts used for the ICL evaluations are provided in Appendix B.7. For ICL TRTS, we use real examples as in-context shots and classify synthetic samples; for ICL TSTR, we reverse this setup and use synthetic shots to classify real samples.

Importantly, higher scores on BERT TRTS, ICL tasks, or text quality metrics should not be interpreted as strictly higher fidelity to the real-data distribution. Even the real-data baseline does not necessarily achieve high values on these metrics, as many datasets are not cleanly separable along a single axis. For example, a Yelp review may exhibit mixed sentiment, and a BioRxiv abstract may span multiple subdomains. In contrast, several baselines attain TRTS scores substantially higher than real data, which suggests that their generations may be *overly label-obvious* (i.e., trivially separable) rather than reflecting the nuanced structure of the underlying corpus.

## D.2  Temperature

In Figure 8, we examine the effect of temperature on all reported metrics. We run LLAMA-3.1-8B on IMDb at $\varepsilon = 5$. At low temperatures, generations are conservative and repetitive, yielding limited lexical and stylistic diversity relative to the real corpus. Increasing the temperature initially improves coverage and distributional fidelity, leading to closer performance on metrics with real data. However, at sufficiently high temperatures, sampling noise dominates and the model produces off-distribution content, causing fidelity to degrade.

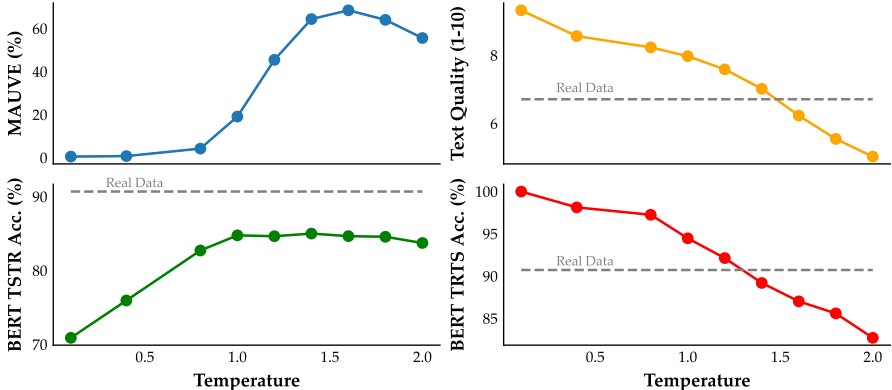

*Figure 8.* Effects of temperature on PT models on IMDb with $\varepsilon = 5$.

## D.3  Layer and Injection Coefficient

Figure 9 studies the sensitivity of EPSVEC to the injection coefficient $\beta$ and the choice of injection layers. We sweep $\beta$ over a range of values and inject the dataset vector into four contiguous layer blocks, while keeping all other settings fixed (backbone: pretrained LLAMA-3.1-8B, dataset: Yelp, $\varepsilon = 5$, and the same decoding/evaluation protocol).

Overall, performance is most stable for moderate injection strength and for injections applied in later layers. Increasing $\beta$ initially improves fidelity (MAUVE) and text quality, indicating that stronger steering can better move generations toward the target distribution. However, overly large $\beta$ degrades multiple metrics, consistent with over-steering that pushes generations off-distribution or reduces semantic consistency. Across layer choices, later-layer injection tends to be more robust, whereas earlier-layer injection is more sensitive to $\beta$ and can lead to sharper degradation at high coefficients. Based on this sweep, we select a middle-range $\beta$ and a late-layer injection block in the main experiments.

Empirically, using too few layers can make the intervention too weak, while injecting into too many layers can make steering less stable. In addition, releasing more per-layer vectors is not privacy-efficient: Algorithm 1 releases one privatized vector per layer, and Theorem 4.1 shows that privacy composes across layers by basic composition, so increasing the number of injected layers increases the total privacy cost (or, equivalently, requires more noise for a fixed privacy budget). Earlier layers are less useful for the high-level semantic and stylistic attributes that EPSVECaims to steer, therefore, we focus on a small block of later layers rather than spreading the intervention across the entire network.

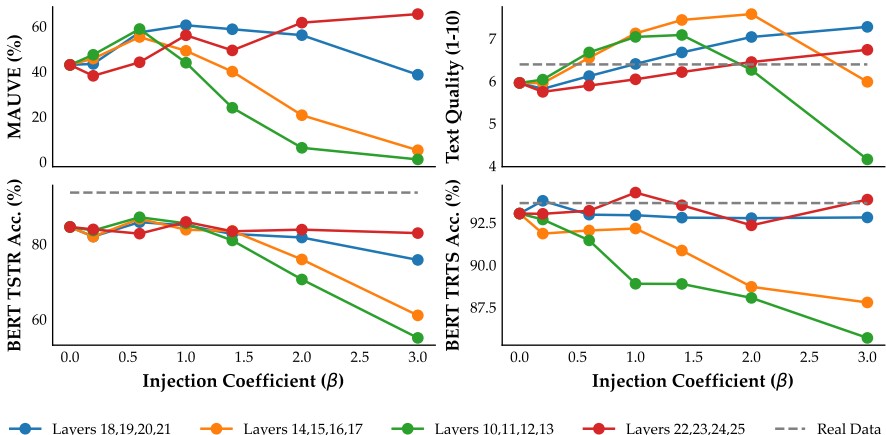

*Figure 9.* Effects of injection layers and coefficient on performance on PT models on Yelp with $\varepsilon = 5$.

## D.4  Multiple Real Shots at Vector Extraction

We directly evaluate whether using multiple real shots during vector extraction improves the quality of dataset vectors. Concretely, we compare the default EPSVEC++ setup against a variant that uses two real shots during vector extraction. Results on the IMDb dataset are shown in Table 6.

Overall, the results do not show a consistent advantage from using additional real shots. At $\varepsilon = 5$, the default EPSVEC++ setup outperforms the 2-shot variant on both MAUVE (72.3 vs. 64.6) and BERT accuracy (84.3 vs. 82.0). At $\varepsilon = 3$, using two shots slightly improves MAUVE (70.2 vs. 67.8), but lowers BERT accuracy (83.3 vs. 84.7). These results suggest that additional real shots during vector extraction may sometimes improve fidelity in tighter privacy regimes, but do not provide a robust improvement across settings. We therefore view the number of real shots as a tradeoff-dependent design choice rather than a universally better variant.

| $\varepsilon$ | Method | Model Type | MAUVE | BERT |
|---|---|---|---|---|
| 5 | EPSVEC++ with 2-shots at vector extraction | PT | $64.6 \pm 0.0$ | $82.0 \pm 0.0$ |
| 5 | EPSVEC++ | PT | $72.3 \pm 2.7$ | $84.3 \pm 0.5$ |
| 3 | EPSVEC++ with 2-shots at vector extraction | PT | $70.2 \pm 0.0$ | $83.3 \pm 0.0$ |
| 3 | EPSVEC++ | PT | $67.8 \pm 1.6$ | $84.7 \pm 1.6$ |

*Table 6.* Effect of using multiple real shots during vector extraction on IMDb.

## D.5  Token Position for Dataset Vector Extraction

In our main experiments, we compute datapoint representations by averaging hidden states across all token positions, as described in Equation (1). Empirically, we found that mean pooling across tokens is more stable and achieves stronger overall performance.

To further evaluate this design choice, we compare the default EPSVEC++ setup against a variant that extracts dataset vectors using only the final token representation. Results on the IMDb dataset are reported in Table 7.

Overall, using the final token still yields reasonable performance, but averaging across all token positions consistently achieves stronger or more stable fidelity. At $\varepsilon = 5$, the default setup improves MAUVE from 66.5 to 72.3, while the last-token variant attains slightly higher BERT accuracy. At $\varepsilon = 3$, the default setup improves both MAUVE and BERT accuracy. These results support our choice of mean pooling as the default representation strategy.

## D.6  Rejection Sampling

Table 8 reports the impact of applying rejection sampling using an LLM-as-a-judge text-quality score (threshold 6) as a post-processing step. We compare EPSVEC++ with and without rejection sampling on the same setting (PT model, $\varepsilon = 5$) as the main table.

Rejection sampling increases the average text-quality score for EPSVEC++ from 5.9 to 6.2 while also improving MAUVE

| $\varepsilon$ | Method | Model Type | MAUVE | BERT |
|---|---|---|---|---|
| 5 | EPSVEC++ | PT | $72.3 \pm 2.7$ | $84.3 \pm 0.5$ |
| 5 | EPSVEC++ last token | PT | $66.5 \pm 0.0$ | $87.6 \pm 0.0$ |
| 3 | EPSVEC++ | PT | $67.8 \pm 1.6$ | $84.7 \pm 1.6$ |
| 3 | EPSVEC++ last token | PT | $65.0 \pm 0.0$ | $83.4 \pm 0.0$ |

*Table 7.* Effect of token position used for dataset vector extraction on IMDb.

$(69.4 \rightarrow 72.3)$ and BERT accuracy $(83.9 \rightarrow 84.3)$. This suggests that, in our setting, enforcing a minimum quality constraint can improve surface-level fluency without sacrificing—and in fact improving—distributional fidelity. Nevertheless, given that gains are moderate, we treat rejection sampling as an optional post-processing step rather than a core component.

| $\varepsilon$ | Baseline | Model | MAUVE | BERT | Text Quality |
|---|---|---|---|---|---|
| $\infty$ | Real Data | - | $89.3_{\pm 1.7}$ | $90.7_{\pm 0.1}$ | $6.7_{\pm 0.0}$ |
| $\infty$ | 2-Shots | PT | $61.3_{\pm 0.4}$ | $87.4_{\pm 0.1}$ | $5.6_{\pm 0.0}$ |
| 5.0 | EPSVEC++ w/o Rejection Sampling | PT | $69.4_{\pm 4.0}$ | $83.9_{\pm 1.3}$ | $5.9_{\pm 0.0}$ |
| 5.0 | EPSVEC++ | PT | $72.3_{\pm 2.7}$ | $84.3_{\pm 0.5}$ | $6.2_{\pm 0.0}$ |

*Table 8.* Effect of rejection sampling with text quality threshold 6. We generate more synthetic samples and then filter samples by using LLM-as-a-judge.

| $\varepsilon$ | Method | Yelp Reviews | | | | | |
|---|---|---|---|---|---|---|---|
| | | MAUVE (%) ↑ | BERT TRTS (%) | BERT TSTR (%) ↑ | ICL TRTS(%) | ICL TSTR(%) | Quality (1-10) |
| $\infty$ | 2-Shot | $66.9_{\pm2.6}$ | $89.7_{\pm0.8}$ | $91.5_{\pm0.1}$ | $91.2_{\pm0.4}$ | $95.3_{\pm0.2}$ | $5.9_{\pm0.0}$ |
| | Real Data | $95.9_{\pm1.2}$ | $93.7_{\pm0.4}$ | $93.6_{\pm0.4}$ | $96.0_{\pm0.4}$ | $96.0_{\pm0.2}$ | $6.4_{\pm0.0}$ |
| 5 | AugPE | $0.4_{\pm0.0}$ | $97.9_{\pm1.0}$ | $81.6_{\pm3.4}$ | $98.6_{\pm0.8}$ | $95.6_{\pm0.1}$ | $7.8_{\pm0.1}$ |
| | InvisibleInk | $0.5_{\pm0.0}$ | $99.7_{\pm0.1}$ | $88.5_{\pm0.5}$ | $99.7_{\pm0.1}$ | $95.3_{\pm0.2}$ | $9.4_{\pm0.0}$ |
| | PP | $8.9_{\pm0.5}$ | $98.2_{\pm0.2}$ | $91.0_{\pm0.3}$ | $98.8_{\pm0.3}$ | $95.8_{\pm0.2}$ | $8.0_{\pm0.0}$ |
| | PP++ | $69.8_{\pm3.0}$ | $83.4_{\pm0.2}$ | $91.3_{\pm0.4}$ | $84.9_{\pm0.6}$ | $96.1_{\pm0.1}$ | $4.9_{\pm0.0}$ |
| | EPSVEC | $12.8_{\pm2.2}$ | $91.7_{\pm1.1}$ | $75.8_{\pm3.6}$ | $96.3_{\pm0.7}$ | $95.1_{\pm0.3}$ | $6.7_{\pm0.1}$ |
| | EPSVEC++ | $62.9_{\pm4.0}$ | $92.9_{\pm0.1}$ | $83.8_{\pm1.2}$ | $93.7_{\pm0.2}$ | $95.7_{\pm0.4}$ | $6.7_{\pm0.1}$ |
| 3 | AugPE | $0.4_{\pm0.0}$ | $97.8_{\pm0.8}$ | $82.5_{\pm1.4}$ | $98.4_{\pm0.9}$ | $95.8_{\pm0.2}$ | $7.8_{\pm0.1}$ |
| | InvisibleInk | $0.5_{\pm0.0}$ | $99.6_{\pm0.0}$ | $87.1_{\pm0.3}$ | $99.5_{\pm0.1}$ | $95.5_{\pm0.2}$ | $9.4_{\pm0.0}$ |
| | PP | $9.1_{\pm0.7}$ | $98.3_{\pm0.1}$ | $90.8_{\pm0.2}$ | $98.7_{\pm0.2}$ | $95.7_{\pm0.2}$ | $8.0_{\pm0.0}$ |
| | PP++† | - | - | - | - | - | - |
| | EPSVEC | $12.2_{\pm1.1}$ | $93.0_{\pm0.6}$ | $76.9_{\pm2.6}$ | $96.9_{\pm0.8}$ | $95.3_{\pm0.3}$ | $6.7_{\pm0.1}$ |
| | EPSVEC++ | $67.3_{\pm3.6}$ | $92.6_{\pm1.2}$ | $85.8_{\pm1.4}$ | $94.1_{\pm0.8}$ | $95.4_{\pm0.3}$ | $6.6_{\pm0.1}$ |

| $\varepsilon$ | Method | IMDb Reviews | | | | | |
|---|---|---|---|---|---|---|---|
| | | MAUVE (%) ↑ | BERT TRTS (%) | BERT TSTR (%) ↑ | ICL TRTS(%) | ICL TSTR(%) | Quality (1-10) |
| $\infty$ | 2-Shot | $61.3_{\pm0.4}$ | $81.2_{\pm0.7}$ | $87.4_{\pm0.1}$ | $84.3_{\pm0.4}$ | $93.3_{\pm0.6}$ | $5.6_{\pm0.0}$ |
| | Real Data | $89.3_{\pm1.7}$ | $90.7_{\pm0.1}$ | $90.7_{\pm0.1}$ | $93.6_{\pm0.4}$ | $93.2_{\pm0.7}$ | $6.7_{\pm0.0}$ |
| 5 | AugPE | $0.5_{\pm0.0}$ | $97.4_{\pm0.4}$ | $73.1_{\pm1.6}$ | $98.0_{\pm0.8}$ | $94.3_{\pm0.4}$ | $8.1_{\pm0.1}$ |
| | InvisibleInk | $0.4_{\pm0.0}$ | $99.5_{\pm0.2}$ | $78.7_{\pm2.0}$ | $99.6_{\pm0.2}$ | $93.5_{\pm0.7}$ | $9.3_{\pm0.0}$ |
| | PP | $3.1_{\pm0.4}$ | $93.9_{\pm1.4}$ | $57.4_{\pm1.3}$ | $92.7_{\pm0.7}$ | $93.5_{\pm0.4}$ | $7.3_{\pm0.1}$ |
| | PP++ | $14.9_{\pm2.2}$ | $75.8_{\pm3.5}$ | $54.2_{\pm0.3}$ | $73.6_{\pm3.6}$ | $93.9_{\pm0.6}$ | $4.5_{\pm0.2}$ |
| | EPSVEC | $8.4_{\pm1.6}$ | $96.2_{\pm0.4}$ | $86.9_{\pm0.6}$ | $96.9_{\pm0.8}$ | $93.7_{\pm0.7}$ | $5.2_{\pm0.1}$ |
| | EPSVEC++ | $72.3_{\pm2.7}$ | $86.9_{\pm0.4}$ | $84.3_{\pm0.5}$ | $89.0_{\pm0.2}$ | $93.3_{\pm0.6}$ | $6.2_{\pm0.0}$ |
| 3 | AugPE | $0.5_{\pm0.0}$ | $97.4_{\pm0.3}$ | $74.3_{\pm3.4}$ | $98.1_{\pm0.8}$ | $94.5_{\pm0.3}$ | $8.1_{\pm0.1}$ |
| | InvisibleInk | $0.4_{\pm0.0}$ | $99.4_{\pm0.3}$ | $77.7_{\pm2.4}$ | $99.6_{\pm0.1}$ | $93.5_{\pm0.7}$ | $9.3_{\pm0.0}$ |
| | PP | $3.1_{\pm0.4}$ | $94.0_{\pm0.9}$ | $57.4_{\pm1.3}$ | $92.7_{\pm0.7}$ | $93.5_{\pm0.4}$ | $7.3_{\pm0.1}$ |
| | PP++† | - | - | - | - | - | - |
| | EPSVEC | $7.8_{\pm0.9}$ | $96.6_{\pm0.2}$ | $86.8_{\pm0.4}$ | $97.2_{\pm0.4}$ | $93.7_{\pm0.8}$ | $5.2_{\pm0.1}$ |
| | EPSVEC++ | $67.8_{\pm1.6}$ | $87.2_{\pm1.2}$ | $84.7_{\pm1.6}$ | $89.2_{\pm0.2}$ | $93.7_{\pm0.8}$ | $6.2_{\pm0.0}$ |

| $\varepsilon$ | Method | BioRxiv Abstracts | | | | | |
|---|---|---|---|---|---|---|---|
| | | MAUVE (%) ↑ | BERT TRTS (%) | BERT TSTR (%) ↑ | ICL TRTS(%) | ICL TSTR(%) | Quality (1-10) |
| $\infty$ | 2-Shot | $76.9_{\pm3.4}$ | $83.1_{\pm0.4}$ | $87.4_{\pm0.7}$ | $68.8_{\pm0.3}$ | $77.1_{\pm0.8}$ | $6.9_{\pm0.0}$ |
| | Real Data | $96.6_{\pm0.6}$ | $91.8_{\pm0.6}$ | $91.8_{\pm0.4}$ | $77.7_{\pm0.3}$ | $77.8_{\pm0.5}$ | $8.8_{\pm0.0}$ |
| 5 | AugPE | $0.5_{\pm0.0}$ | $25.3_{\pm0.4}$ | $23.8_{\pm1.0}$ | $21.5_{\pm0.5}$ | $76.8_{\pm0.7}$ | $6.7_{\pm0.0}$ |
| | InvisibleInk | $0.9_{\pm0.0}$ | $94.6_{\pm0.7}$ | $86.3_{\pm0.7}$ | $89.2_{\pm0.8}$ | $74.2_{\pm0.2}$ | $9.4_{\pm0.0}$ |
| | PP | $1.7_{\pm0.1}$ | $96.8_{\pm0.8}$ | $26.9_{\pm0.5}$ | $89.6_{\pm4.5}$ | $73.0_{\pm0.5}$ | $7.0_{\pm0.1}$ |
| | PP++ | $1.9_{\pm0.1}$ | $75.2_{\pm5.0}$ | $26.7_{\pm0.7}$ | $59.2_{\pm7.7}$ | $76.6_{\pm0.3}$ | $5.5_{\pm0.1}$ |
| | EPSVEC | $35.8_{\pm0.9}$ | $96.0_{\pm0.2}$ | $86.4_{\pm0.6}$ | $90.9_{\pm0.2}$ | $76.5_{\pm0.4}$ | $5.9_{\pm0.0}$ |
| | EPSVEC++ | $62.2_{\pm0.2}$ | $90.6_{\pm0.4}$ | $86.0_{\pm1.9}$ | $80.2_{\pm0.6}$ | $76.8_{\pm0.4}$ | $7.6_{\pm0.0}$ |
| 3 | AugPE | $0.5_{\pm0.0}$ | $25.5_{\pm0.4}$ | $22.7_{\pm0.7}$ | $21.4_{\pm0.5}$ | $76.8_{\pm0.3}$ | $6.7_{\pm0.0}$ |
| | InvisibleInk | $0.9_{\pm0.0}$ | $94.4_{\pm0.5}$ | $85.6_{\pm0.9}$ | $89.3_{\pm0.4}$ | $74.0_{\pm0.5}$ | $9.4_{\pm0.0}$ |
| | PP | $1.7_{\pm0.1}$ | $97.4_{\pm0.8}$ | $26.9_{\pm0.5}$ | $89.6_{\pm4.5}$ | $73.0_{\pm0.5}$ | $7.0_{\pm0.1}$ |
| | PP++† | - | - | - | - | - | - |
| | EPSVEC | $37.2_{\pm1.2}$ | $95.3_{\pm0.3}$ | $85.3_{\pm0.2}$ | $90.4_{\pm0.6}$ | $76.4_{\pm0.5}$ | $6.0_{\pm0.0}$ |
| | EPSVEC++ | $60.7_{\pm1.1}$ | $89.8_{\pm0.5}$ | $85.8_{\pm1.7}$ | $78.8_{\pm0.5}$ | $77.8_{\pm0.5}$ | $7.6_{\pm0.0}$ |

| $\varepsilon$ | Method | ICLR Reviews on OpenReview | | | | | |
|---|---|---|---|---|---|---|---|
| | | MAUVE (%) ↑ | BERT TRTS (%) | BERT TSTR (%) ↑ | ICL TRTS(%) | ICL TSTR(%) | Quality (1-10) |
| $\infty$ | 2-Shot | $56.8_{\pm0.7}$ | $68.3_{\pm1.1}$ | $69.9_{\pm0.8}$ | $69.1_{\pm0.6}$ | $71.6_{\pm0.5}$ | $6.1_{\pm0.0}$ |
| | Real Data | $95.8_{\pm0.6}$ | $73.1_{\pm0.1}$ | $73.2_{\pm0.1}$ | $71.7_{\pm0.5}$ | $72.0_{\pm0.2}$ | $7.1_{\pm0.0}$ |
| 5 | AugPE | $0.4_{\pm0.0}$ | $50.7_{\pm0.1}$ | $50.9_{\pm0.6}$ | $50.3_{\pm0.3}$ | $74.2_{\pm0.4}$ | $7.8_{\pm0.0}$ |
| | InvisibleInk | $0.5_{\pm0.0}$ | $97.9_{\pm0.2}$ | $61.7_{\pm1.1}$ | $98.7_{\pm0.1}$ | $72.0_{\pm0.3}$ | $9.1_{\pm0.0}$ |
| | PP | $2.3_{\pm0.6}$ | $76.9_{\pm4.3}$ | $50.0_{\pm0.2}$ | $80.7_{\pm3.4}$ | $73.5_{\pm0.6}$ | $6.7_{\pm0.1}$ |
| | PP++ | $5.5_{\pm1.3}$ | $65.0_{\pm5.0}$ | $49.9_{\pm0.1}$ | $62.5_{\pm2.9}$ | $74.2_{\pm0.4}$ | $4.8_{\pm0.1}$ |
| | EPSVEC | $11.9_{\pm0.3}$ | $92.3_{\pm0.8}$ | $67.6_{\pm0.4}$ | $94.1_{\pm0.3}$ | $73.1_{\pm0.2}$ | $5.2_{\pm0.1}$ |
| | EPSVEC++ | $33.0_{\pm1.0}$ | $78.7_{\pm0.4}$ | $66.4_{\pm0.8}$ | $83.9_{\pm0.7}$ | $73.4_{\pm0.6}$ | $5.9_{\pm0.1}$ |
| 3 | AugPE | $0.4_{\pm0.0}$ | $50.7_{\pm0.2}$ | $50.6_{\pm0.9}$ | $50.3_{\pm0.2}$ | $73.9_{\pm0.3}$ | $7.8_{\pm0.0}$ |
| | InvisibleInk | $0.4_{\pm0.0}$ | $98.3_{\pm0.1}$ | $62.8_{\pm0.7}$ | $99.3_{\pm0.2}$ | $71.5_{\pm0.5}$ | $9.2_{\pm0.0}$ |
| | PP | $2.3_{\pm0.6}$ | $76.2_{\pm4.4}$ | $50.0_{\pm0.2}$ | $80.7_{\pm3.4}$ | $73.5_{\pm0.6}$ | $6.7_{\pm0.1}$ |
| | PP++† | - | - | - | - | - | - |
| | EPSVEC | $11.1_{\pm0.1}$ | $92.7_{\pm0.6}$ | $68.2_{\pm1.4}$ | $95.4_{\pm0.5}$ | $73.2_{\pm0.6}$ | $5.3_{\pm0.0}$ |
| | EPSVEC++ | $33.0_{\pm1.8}$ | $80.4_{\pm0.1}$ | $67.3_{\pm0.5}$ | $84.4_{\pm0.2}$ | $73.5_{\pm0.0}$ | $5.9_{\pm0.0}$ |

*Table 9.* Complete results of Table 2, with the addition of text quality score, BERT TRTS, and in-context learning. Note that higher BERT TRTS, ICL, and text quality score do not necessarily imply that synthetic data is closer to real data.

# E   Limitations

Dataset vectors rely on the separability between public and private data. While we observe this separability across multiple text domains and widely-used datasets, it is not a universal guarantee for all datasets. Therefore, it remains unclear whether our method can be applied to more complex datasets like high-dimensional biological data.

Moreover, dataset vectors are most effective when the model can already provide a reasonable fixed-shot candidate pool. In other words, our method excels in domains where the model has previously seen the data or where public data is available. For domains where the data is fully novel without any prior data release, the lack of public data can reduce the effectiveness of our method.

Finally, as is common in private synthetic data generation, we use MAUVE to quantify distributional similarity between synthetic and real text. While MAUVE is a useful distributional proxy, it does not directly measure semantic faithfulness, factual consistency, or coherence. This limitation is especially salient for domains with richer semantic structure, where distributional metrics may not reliably detect. Developing evaluation metrics that are less sensitive to hyperparameters and more aligned with semantic correctness remains an important direction for future work.

# F   Final Discussion

This work shows that EPSVEC can make training-free private text generation both practical and high-fidelity by shifting the unit of privatization from tokens to a compact representation of dataset shift. Rather than privatizing each sample, EPSVEC privatizes a single dataset-level direction that captures how generations should move toward the target corpus. Concretely, EPSVEC extracts dataset vectors from private embeddings, releases a single sanitized vector satisfying differential privacy, and then reuses it for all downstream decoding and post-processing at no additional privacy cost. To our knowledge, this is the first DP synthetic data pipeline that operationalizes steering vectors as a reusable, privatized control signal for dataset-level distributional alignment. This immediately yields an appealing operating point: arbitrary numbers of samples, arbitrary sequence lengths, and inference efficiency comparable to standard generation rather than per-token DP aggregation.

Empirically, EPSVEC delivers strong distributional fidelity across four corpora, with especially large gains in lower-data domains (IMDb, BioRxiv, OpenReview). In aggregate, the method achieves large fidelity improvements while maintaining competitive downstream utility when synthetic data are used for BERT fine-tuning. These results suggest that a single privatized direction can encode rich, high-dimensional dataset attributes (style, subtopic mixtures, semantic flow) that are difficult to match via prompting alone, especially under tight privacy budgets.

Moreover, EPSVEC meaningfully improves deployability and scalability. Inference-time aggregation baselines face a scalability bottleneck because they must aggregate token distributions over large private batches or requires more samples and compute for effective generations, making long-form generation and large-scale sampling expensive; in experiments, this prevents certain baselines from producing the full 2K samples on several corpora and at stronger privacy. By contrast, EPSVEC attains strong fidelity while requiring the least compute and relatively few private samples, and it remains viable for longer sequences under tight budgets. This combination of one-time privatization and unconstrained downstream generation represents a qualitatively different scalability regime from prior training-free DP approaches.

Finally, the fixed-shot exemplar component highlights an important practical lesson: pretrained (base) models often provide broader stylistic support than instruction-tuned variants, but they can be harder to stabilize with zero-shot prompting. We emphasize that our contribution is not the first use of pretrained models for private synthetic generation; prior work has also leveraged PT models, often by directly incorporating real data samples into prompts. Instead, our contribution is a practical and stable pipeline that replaces direct real-data prompting with reusable DP fixed-shots, enabling pretrained models to work effectively in the DP steering setting. In particular, privately selecting a small set of fixed-shot exemplars via DP histograms provides a reusable scaffold that anchors both dataset-vector construction and downstream generation, substantially improving controllability and fidelity. Looking forward, the observed non-monotonic privacy–utility behavior (where moderate noise can improve MAUVE) is an intriguing signal that privacy noise may sometimes act as a useful regularizer, suggesting a promising direction for future work.

