# OpenReview forum: "EPSVec: Efficient and Private Synthetic Data Generation via Dataset Vectors"
_ICML.cc/2026/Conference — ICML 2026 regular_

### Official Review · Reviewer_oUMp · 2026-03-09

**Soundness:** 3
**Presentation:** 3
**Significance:** 3
**Originality:** 3
**Overall Recommendation:** 5
**Confidence:** 4

**Summary:**

The paper proposes a novel way to generate textual synthetic data based on real data with differential privacy guarantees. Specifically, they demonstrate how dataset vectors (that measure the distribution shifts between synthetic and private real data) can be used to improve the generated synthetic data quality while retaining the privacy of the real dataset. This is further enhanced through use of fixed-shot examples to improve diversity and fidelity of data by using some of the privacy budget. The claims are supported on experiments that evaluate the generated data quality on the data similarity (using MAUVE) and downstream performance (using BERT metrics).

**Compliance With Llm Reviewing Policy:**

Affirmed.

**Key Questions For Authors:**

The private and synthetic sets have paired samples (Algorithm 1). Is the pairing arbitrary? Is there some form of sample selection? What happens if the paired samples are very different and there is limited meaningful comparison between them?

During downstream task performance evaluation, the tasks evaluated on are very coarse level, focusing on distinguishing between broad categories. Could this approach be used for per sentence level annotations? It seems to rely on per document level annotation.

**Limitations:**

The authors discuss limitations in privacy-utility trade-off to do with privacy budget.

Further, the degradation of accuracy in terms of improved diversity is also discussed.

**Strengths And Weaknesses:**

Paper strengths:
- The paper is clearly written and easy to follow. The algorithm and method are clearly presented and justified.
- The evaluation is done both on data similarity and on downstream task performance
- Interesting analysis of diversity collapse and analysing of synthetic data beyond single sample quality but rather overall dataset quality
- Theoretical proof of differential privacy guarantees when using the approach
- Quite a few ablations demonstrating importance of different components to overall performance

Weaknesses:
- The private and synthetic sets have paired samples (Algorithm 1). Is the pairing arbitrary? Is there some form of sample selection? What happens if the paired samples are very different and there is limited meaningful comparison between them?
- During downstream task performance evaluation, the tasks evaluated on are very coarse level, focusing on distinguishing between broad categories. Could this approach be used for per sentence level annotations? It seems to rely on per document level annotation.

---

> ### Author Rebuttal · Authors · 2026-03-30
>
> # Response to Reviewer 6xDB
> We thank the reviewer for the positive assessment of the paper’s clarity, theory, and ablation analysis.
>
> ## Pairing samples in Algorithm 1
> The construction does not rely on semantic one-to-one alignment between private and synthetic records. For each attribute $y$, we form a private set $\mathcal{D}^+_y$ and a synthetic reference set $\mathcal{D}^-_y$, and in practice pair samples randomly within that attribute-conditioned subset; we do not use any additional matching or sample-selection procedure. Importantly, in the unclipped setting, the exact pairing does not affect the final mean-difference estimate, since the average of pairwise differences is just the difference of the empirical means. The main requirement is therefore that both sets share the same high-level attribute $y$, not that individual examples be closely aligned. If the reference set is poorly matched, the estimate can become noisier, which is why fixed-shots are used to improve $\mathcal{D}^-_y$ before vector construction.
>
> ## Per-sample annotations
> We agree that the current experiments focus on document-level labels. Conceptually, EPSVec is defined over labeled text units rather than specifically over documents, so the same framework could in principle be applied to sentence-level classification by constructing positive and negative sets over sentences instead of documents.
> Importantly, from a privacy perspective, this extension is not directly clear. Moving from document-level to sentence-level units may change the privacy regime substantially: shorter spans can contain less contextual averaging, may be more sensitive to unique phrasing, and may require different assumptions about what constitutes one protected record. As a result, while the EPSVec construction is flexible at the modeling level, its privacy-utility behavior for sentence-level annotation would need to be studied separately.

---

> > ### Author Rebuttal · Reviewer_oUMp · 2026-04-01
> >
> > Concerns addressed

---

> > > ### Author Response · Authors · 2026-04-04
> > >
> > > Thank you for the update. We’re glad the clarification addressed your concerns, and are grateful for your constructive suggestions!

---

### Official Review · Reviewer_6xDB · 2026-03-09

**Soundness:** 2
**Presentation:** 3
**Significance:** 3
**Originality:** 2
**Overall Recommendation:** 4
**Confidence:** 4

**Summary:**

This paper proposes a differentially-private lightweight framework to utilize LLMs for synthetic data generation, i.e., EPSVEC, which aims to efficiently generate high-quality synthetic data with rigorous privacy guarantee. Specifically, EPSVEC extract distilled private dataset vector which could compactly encode a dataset-level characteristic properties, and release a perturbed version of the vector that satisfies differential privacy. By injecting them to the hidden states of the LLM at inference time, to guide the decoding process of LLMs in order to generate high-quality synthetic datasets.  The authors present extensive experimental evaluations of EPSVEC. The results indicate that EPSVEC achieves a 150% fidelity (i.e., MAUVE) improvement across four datasets compared to the second-best baseline.

**Compliance With Llm Reviewing Policy:**

Affirmed.

**Final Justification:**

The author's response has addressed most of my concerns, and I will keep the positive score for the paper.

**Key Questions For Authors:**

1)	Steering LLMs in the embedding space presents a promising direction. Consequently, one of the most immediate concerns is how to compute the dataset vectors. And in Section 5.1, Hyperparameters, 4 layers (layer 18-21) are used to extract dataset vectors, Why were four layers chosen, and why are these specific four layers selected for this model? (It is noted in Appendix D that different models may require different layers.) Is there a reasonable mechanism behind the selection of these four layers? Also, could the choice of different layers lead to instability in the method’s performance?

2)	In Table 2, even with the best-performing EPSVEC, PT model and IT model exhibit clear differences in their strengths on metrics MAUVE and BERT: PT model performs better on metric fidelity, while IT model excels on downstream performance. Is there any deeper analysis regarding the possible reasons behind this, as well as suggestions on how to balance or prioritize these two metrics?

**Limitations:**

The limitation was not discussed in paper, but in appendix.

**Strengths And Weaknesses:**

Strengths:
1)	The pipeline requires only a one-time release of private dataset vectors, according to the differential privacy post-processing theorem, does not incur any additional privacy cost. Therefore, this mechanism avoids the high privacy overhead associated with per-token or per-sample approaches.
2)	This paper considers the different capabilities of pre-trained (PT) base models and instruction-tuned (IT) models in terms of generating diverse and qualified synthetic data, which is an interesting and often overlooked perspective. With proposed fixed-shot prompt templates, they manage to utilize PT model to achieve higher synthetic
3)	Extensive experiments are constructed to compare EPSVEC with three different baselines over four different datasets, the results show that EPSVEC marginally outperform baselines in terms of distributional fidelity, i.e., MAUVE.
4)	The paper is clearly written and logically structured, and the method design is based on sound findings from preliminary studies.

Weaknesses:
1)	The analysis of Table 2 is not sufficiently comprehensive. Although two metrics are introduced, only MAUVE is emphasized in the analysis, and downstream performance is not discussed in detail. While MAUVE is certainly an important metric, downstream performance is equally crucial for evaluating the utility of synthetic data in practical applications.
2)	The method is evaluated on relatively short text datasets; however, its scalability to long and complex real-world private text datasets (such as medical records or financial reports containing more than 5,000 tokens, not public) remains unclear, both in terms of data quality and efficiency.
3)	There is inconsistency in the wording, in Table. 2, caption, it presents methods with ++ use pretrained (PT) models, but in Section 5.2, at the end of paragraph two, it says: “pretrained vs. instruction-tuned models (PP vs. PP++)”. This is conflict.

---

> ### Author Rebuttal · Authors · 2026-03-30
>
> # Response to Reviewer 6xDB
>
> We thank the reviewer for the careful reading and for highlighting the overall clarity of the paper. We appreciate the concrete suggestions provided.
>
>
> ## Evaluation on downstream tasks
>
> We agree that downstream utility is important, which is the reason that we report BERT TSTR, and further report results including BERT TRTS, ICL TRTS/TSTR, and text quality in Appendix D.1. **We intentionally avoid equating downstream separability with fidelity as some baselines exceed real data on TRTS-like metrics due to overly label-obvious text, rather than text that faithfully matches the real data, as noted in Appendix D.1.** Our intended claim is therefore balanced: EPSVec provides the strongest fidelity and competitive downstream utility, rather than optimizing only one metric. We will make this distinction more explicit in the discussion of Table 2.
>
>
> ## High sequence length and specialized domains
>
> We also agree that evaluation on highly specialized long-form private corpora is an important direction. However, EPSVec changes the privacy/computation regime by privatizing a single dataset-level direction once, after which generation and post-processing incur no additional privacy cost. **This supports arbitrary sample counts and sequence lengths at standard-generation efficiency, in contrast to inference-time aggregation baselines that must aggregate token distributions over large private batches and, in our experiments, fail to produce the full 2K samples on several corpora/settings.** Whether fidelity remains equally strong on highly specialized domains such as medical or financial reports is a separate empirical question that we do not intend to overclaim.
>
>
> ## Inconsistency in wording
>
> Thank you for catching the inconsistencies in our wording. Our intended meaning is that in Table 2, the ‘++’ variants denote the pretrained-model versions, whereas the non-++ variants use instruction-tuned models; we will revise Section 5.2 to make this unambiguous.
>
> ## Answers to reviewer’s questions
>
> ### Q1: Why 4 layers? Why layers 18-21? Is the method stable across layer choice?
>
> Regarding the layer choice: for LLAMA-3.1-8B we use layers 18–21 with coefficient 1.4, chosen empirically based on the ablations in Appendix D. Importantly, the optimal injection region is model-dependent, which is why Table 5 reports different layer ranges for different backbones. We also note that Appendix D shows that other layer choices can still yield reasonable performance when paired with an appropriate injection coefficient. In practice, however, very large coefficients can make steering unstable, while very small coefficients are often ineffective.
>
> Why four layers? Empirically, using too few layers can make the intervention too weak, while injecting into too many layers can make steering less stable. In addition, releasing more per-layer vectors is not privacy-efficient: Algorithm 1 releases one privatized vector per layer, and Theorem 4.1 shows that privacy composes across layers by basic composition, so increasing the number of injected layers increases the total privacy cost (or, equivalently, requires more noise for a fixed privacy budget). Our qualitative intuition is that earlier layers are less useful for the high-level semantic and stylistic attributes that EPSVec aims to steer, so we focus on a small block of later layers rather than spreading the intervention across the entire network.
>
> Our qualitative finding is broadly consistent with prior layer-wise probing studies, suggesting that deeper layers tend to encode more abstract and contextual information, and that the specific layer choice varies in different models. In particular, [1] argues that more complex concepts are typically acquired in deeper layers, while [2] finds that LLMs preferentially encode more context knowledge in upper layers.
>
>
>
> ### Q2: How to balance PT vs IT on MAUVE vs. downstream performance?
>
> We thank the reviewer for this thoughtful question. On PT vs. IT models, we believe the metric difference reflects a real tradeoff rather than a contradiction: **PT models provide broader stylistic support and therefore stronger MAUVE/fidelity, while IT models may produce more instruction-aligned and label-separable outputs, which can help downstream classification**. Appendix D.1 already cautions that stronger classification scores need not imply better fidelity. We will clarify this interpretation in the revision.
>
> **References**:
>
> [1] Exploring Concept Depth: How Large Language Models Acquire Knowledge and Concept at Different Layers? Jin et al.
>
> [2] How Large Language Models Encode Context Knowledge? A Layer-Wise Probing Study, Ju et al.

---

> > ### Author Rebuttal · Reviewer_6xDB · 2026-04-01
> >
> > The author's response has addressed most of my concerns, and I would be happy to raise my score.

---

> > > ### Author Response · Authors · 2026-04-04
> > >
> > > Thank you for the update! We’re glad the clarifications addressed your concerns, and we sincerely appreciate your willingness to raise your score. We will incorporate these revisions into the main text to improve clarity and ensure the contributions are clearly presented.

---

### Official Review · Reviewer_21WF · 2026-03-11

**Soundness:** 3
**Presentation:** 4
**Significance:** 4
**Originality:** 4
**Overall Recommendation:** 5
**Confidence:** 5

**Summary:**

In this paper, the authors apply and make a simple adjustment of activation steering vector to the synthesis data generation field, which makes this process more efficient while guaranteeing the privacy of data. Rather than just adopting one technique in one field, this paper provides a new perspective for future research in this area. Some good results in the experiment part are provided. In addition, the authors also point out some limitations, such as applying this approach to some special distribution datasets may have some difficulties.

**Compliance With Llm Reviewing Policy:**

Affirmed.

**Key Questions For Authors:**

See in the weakness

**Limitations:**

yes

**Strengths And Weaknesses:**

Advantages:
- The authors provide a novel use for activation steering. They propose data vector that combines the idea with Differential Privacy for synthetic data generation.

- The reading is very easy, and for some readers who do not even have any knowledge of this, they can follow the logic very clearly. I really like the first figure, and after I read that, I can guess what the authors want to do in the following sections.

- The technical design in this paper is very straightforward and clear. Theoretical analysis in this paper further improves its solidity.

Weaknesses:
- The token position used for extracting the activation steering vector can significantly influence the effectiveness of the intervention. Some prior work directly uses the final token position, while others design specific selection mechanisms [1]. The authors should at least provide a rationale for their choice or include ablation studies to analyze the impact of different token positions.

- Minor: All main experiments are conducted on open-source models with scales below 8B parameters. It would strengthen the paper to include results on models with more diverse scales, particularly larger models, to better demonstrate the robustness of the proposed method.

- While the current study focuses on the text domain, this approach may potentially extend to other multimodal settings. It would be valuable for the authors to discuss how the method could be adapted to such domains. Including preliminary results or outlining possible extensions in a future version could further highlight the generality and novelty of the approach.

[1] Arditi A, Obeso O, Syed A, et al. Refusal in language models is mediated by a single direction, NeurIPS 2024.

---

> ### Author Rebuttal · Authors · 2026-03-30
>
> # Response to Reviewer 21WF
>
> We thank the reviewer for the positive assessment of the paper’s novelty, clarity, and theoretical grounding.
>
> ## Token position for extracting vectors
>
> We appreciate the reviewer for pointing out that more explanation is needed for selecting the token position for extracting dataset vectors. For clarity, we used the average over all token positions, since we empirically observed it to be more stable and have higher performance. Yet EPSVec with vectors extracted from the last token position also shows a reasonable performance, as presented in the table below. We will add clarification regarding these new experiments in our revision.
>
> | $\epsilon$| Method | Model Type | MAUVE | BERT |
> |---|---|---|---:|---:|
> | 5 | EPSVec++ | PT | 72.3 ± 2.7 | 84.3 ± 0.5 |
> | 5 | EPSVec++ last token | PT | 66.5 ± 0.0 | 87.6 ± 0.0 |
> | 3 | EPSVec++ | PT | 67.8 ± 1.6 | 84.7 ± 1.6 |
> | 3 | EPSVec++ last token | PT | 65.0 ± 0.0 | 83.4 ± 0.0 |
>
> ## Experiments on larger models
>
> Given the compute resources, it is difficult to conduct thorough experiments during the rebuttal period, especially because we generate a large body of text for statistically significant results. While running private prediction baselines is totally infeasible with larger models, we will, however, attempt to present reasonable experiments with EPSVec on larger models in the final version of the paper to respond to the reviewer’s suggestion.
>
>
> ## Extending to multi-modality
>
> We appreciate the reviewer’s thoughtful suggestion about multi-modal extensions. This extension is not automatic: it would require suitable multimodal representations, well-matched reference distributions, and modality-suitable evaluation metrics. That being said, vector steering has been proven to be effective in the image domain [1] and using EPSVec framework for extracting and privatizing dataset vectors is a promising future direction.
>
>
> [1] General and Efficient Steering of Unconditional Diffusion, Wang et. al.

---

> > ### Author Rebuttal · Reviewer_21WF · 2026-04-03
> >
> > Concerns addressed. Thanks!

---

> > > ### Author Response · Authors · 2026-04-04
> > >
> > > We are grateful for your thoughtful questions and suggestions for further extensions! We will add your suggestions above to the main text.

---

### Official Review · Reviewer_heD6 · 2026-03-22

**Soundness:** 4
**Presentation:** 3
**Significance:** 4
**Originality:** 3
**Overall Recommendation:** 4
**Confidence:** 4

**Summary:**

The authors propose to use steering vectors to generate DP synthetic data. The steering vectors are formed with difference of means between a few-shot generation prompt using (a) model-generated prompted synthetic data and (b) real private data. Steering vectors are added with some tuned coefficients to every token of the response for generation.

The authors find that using a better few-shot generation prompt allows the steering vector to focus on more subtle differences between prompted data and private data, leading to better results. They use a zero-shot generate + DP histogram filtering approach to get better prompts, which improves results.

**Compliance With Llm Reviewing Policy:**

Affirmed.

**Final Justification:**

I maintain my positive assessment of the paper. The reviewer resolved my concern regarding the framing of the contribution regarding using PT models.

**Key Questions For Authors:**

What is fixed shot (no histogram)? Is it just using a random 2 examples from the zero-shot synthetic data?

Do you expect improvements from multiple real shots in the difference of means steering vectors?

Do you expect the steering vectors to be capable of adapting to more specific qualities in the private data? E.g. a specific output format in the private data that is not part of the prompt.

**Limitations:**

Yes

**Strengths And Weaknesses:**

Strengths:
- The empirical results are strong. The method has good MAUVE scores. Generating a lot of data has no privacy scaling and generating scaling with essentially no overhead compared to standard decoding.
- There is some nice intuition and demonstration regarding how better few shot exemplars for synthetic data generation improves the steering vectors.
- The ablations are insightful

Weaknesses:
- The proposed approach seems to have private evolution like scaling -- good at low epsilon but the ceiling at epsilon=infinity is low. What concerns me therefore is if the proposed approach is only strong at the existing simple evaluations in the literature (e.g. yelp polarity classification) but fails to be useful for harder tasks. Generally, lack of better scaling with larger epsilon is a marker of this. Another piece of evidence for this is that the 2-shot real data baseline is very competitive with all approaches, and we don't expect 2 real samples to be informative enough to capture the richness of a dataset of thousands for any real world task.
- The gaps in representativeness between IT and pretrained models is demonstrated in the cited work Amin et al. (2025). Authors also claim that they "introduce a practical technique for reliably leveraging pretrained-only (base) LLMs for private data generation" the aforementioned work, and also Tang et al. (2024) use pretrained LLMs.
- While fixed shot is a nice improvement, ideally one might want a setup where no prompting is used at all. This is the case in the private prediction baselines.

Overall the my concern is regarding the limitations of such an approach to become useful for more ambitious, real-world synthetic data generation tasks, and a lack of evidence in the present paper for this. The improvements over existing methods on existing evaluations still incline me to accept the paper.

---

> ### Author Rebuttal · Authors · 2026-03-30
>
> # Response to Reviewer heD6
>
> We thank the reviewer for recognizing EPSVec’s strengths and appreciate their thoughtful and constructive feedback.
>
> ## Concern on utility in real-world tasks
>
> We agree that richer and long-form datasets are an important future next step. In our current setup, we also evaluate on BioRxiv abstracts and OpenReview reviews, which already provide a richer context as compared to previously benchmarked datasets. An intentional design of EPSVec is also to achieve significant improvements in lower-data regimes, rather than tackling every high-fidelity setting. Moreover, EPSVec does not rely on a large body of private data for generation at scale. Thus, partitioning and clustering can be applied to dense datasets with more features, and separate dataset vectors can be extracted from the clusters for more granular control.
>
>
> As a clarification, note that the *non-private 2-shot baseline with real data* does not use fixed real samples across all prompts. Rather, at every query to the model, we pick two shots at random from the private set and feed the prompt to the model. Therefore, we don’t expect only two samples to be representative of the entire dataset, rather, we hope that sampling shots at random covers the entire dataset well enough.
> We view the strength of the non-private 2-shot baseline as making the comparison more stringent, not as evidence that two samples fully characterize the data distribution. A central message of our paper is to demonstrate that dataset vectors *can capture dataset-level structure beyond a few prompt samples*, hence match or even exceed the non-private 2-shot baseline. Therefore, in richer contexts, these vectors can capture more subtle dataset-level signals.
>
>
> ## Previous works utilizing PT models
>
> We agree that prior work has used pretrained models. However, they use real data samples directly in their prompt, while our method does not. Instead, we propose the fixed-shots as a replacement. Therefore, our intended claim is *not* that we are the first paper to use PT models for private generation. Our contribution is a *practical pipeline* for stably leveraging PT models to work well in DP steering setting. *We will revise the wording to avoid overclaiming novelty on PT model usage, and instead emphasize our practical integration of PT models with DP fixed-shots in our setting.*
>
> ## Preference for no prompting
>
> The prompt-free approach would indeed be appealing. In our setting, the fixed shots are not merely optional prompting, but a DP mechanism that stabilizes PT generations and sharpens the dataset vector.
>
> ## Answers to Reviewer’s Questions
>
> ### Q1. “What is fixed shot (no histogram)? Is it just using a random 2 examples from the zero-shot synthetic data?”
>
> Yes, they are selected randomly once and fixed entirely in the pipeline.
>
> ### Q2 “Do you expect improvements from multiple real shots in the difference-of-means steering vectors?”
>
> We tested this directly by comparing EPSVec++ with 2 real shots at vector extraction against the default setup. Results on IMDb dataset are added below. The results do not show a consistent advantage from using multiple real shots. At $\epsilon=5$, the default EPSVec++ performs better on both MAUVE (72.3 vs. 64.6) and BERT (84.3 vs. 82.0). At $\epsilon=3$, using 2 shots improves MAUVE slightly (67.8 vs. 70.2) but lowers BERT (84.7 vs. 83.3). Overall, this suggests that additional real shots at vector extraction may sometimes help fidelity in tighter-privacy settings, but do not provide a robust across-the-board benefit. We therefore view this as a possible tradeoff-dependent design choice rather than a universally better variant.
>
>
> | $\epsilon$| Method | Model Type | MAUVE | BERT |
> |---|---|---|---:|---:|
> | 5 | EPSVec++ with 2-shots at vector extraction  | PT | 64.6 ± 0.0 | 82.0 ± 0.0 |
> | 5 | EPSVec++ | PT | 72.3 ± 2.7 | 84.3 ± 0.5 |
> | 3 | EPSVec++ with 2-shots at vector extraction | PT | 70.2 ± 0.0 | 83.3 ± 0.0 |
> | 3 | EPSVec++ | PT | 67.8 ± 1.6 | 84.7 ± 1.6 |
>
>
>
> ### Q3 “Do you expect the steering vectors to be capable of adapting to more specific qualities in the private data?”
>
> We expect this to be possible when the format/style difference is consistently reflected across the private corpus, because the vectors encode dataset-level residuals beyond the prompt. Moreover, since EPSVec does not depend on large dataset sizes, depending on the use case, the private data can be partitioned into sets by the specific target quality. Each dataset vector extracted from these distinct partitions potentially captures more fine-grained properties.
>
> That being said, we acknowledge that targeted evaluation on explicit formatting attributes is future work.

---

> > ### Author Rebuttal · Reviewer_heD6 · 2026-04-03
> >
> > Authors resolved my concerns about the claim of PT vs IT usage.
> > W.r.t. my concerns about the prompt, what I meant was the usage of custom, dataset specific prompt engineering. This is used in PE, not so much in prediction based approach, and is used here.

---

> > > ### Author Response · Authors · 2026-04-05
> > >
> > > We appreciate the reviewer for clarifying their concern. We’d like to note that, as mentioned and ablated in our paper, our dataset specific fixed shot prompting is an *optional* component to further boost EPSVec generation quality. In fact, a key advantage of EPSVec over prompting-free baselines is also our compute and data efficiency.
> > >
> > > Moreover, while these fixed shots are indeed dataset specific (and optional), the *principled approach* to extract them reliably is also a contribution of our paper.
> > >
> > > We hope these clarifications address the reviewers concerns.

---

### Decision · Program_Chairs · 2026-04-30

**Decision:**

Accept (regular)

**Comment:**

This paper proposes a method for differentially private synthetic text generation based on dataset vectors that steer model activations. There is agreement among reviewers that the paper is technically sound, well presented, and has strong empirical results. The ablations and analyses also support of the proposed method.

A primary concern shared by mutliple reviewers is the scope of evaluation: experiments are conducted on relatively short text and moderate scale settings, leaving open questions about performance on more complex, long-form, or real-world datasets. Also the downstream utility is not analyzed as deeply as fidelity metrics.

Other technical questions around design choices such as layer selection, token position, and prompting strategy were largely clarified in the rebuttal by the authors. Also concerns about overclaiming novelty with pretrained models, the role of fixed-shot prompting, and certain implementation details were adequately addressed. Reviewers generally maintained or strengthened their positive assessments after rebuttal.

Despite some limitations in experiments, the paper is likely to be of interest to the ICML community and serve as a useful building block for future work. I therefore recommend acceptance.